# Wnt signaling controls pro-regenerative Collagen XII in functional spinal cord regeneration in zebrafish

Daniel Wehner [1], Themistoklis M. Tsarouchas[1], Andria Michael[1], Christa Haase[2], Gilbert Weidinger [3], Michell M. Reimer[4], Thomas Becker[1] & Catherina G. Becker[1]

The inhibitory extracellular matrix in a spinal lesion site is a major impediment to axonal regeneration in mammals. In contrast, the extracellular matrix in zebrafish allows substantial axon re-growth, leading to recovery of movement. However, little is known about regulation and composition of the growth-promoting extracellular matrix. Here we demonstrate that activity of the Wnt/β-catenin pathway in fibroblast-like cells in the lesion site is pivotal for axon re-growth and functional recovery. Wnt/β-catenin signaling induces expression of col12a1a/b and deposition of Collagen XII, which is necessary for axons to actively navigate the non-neural lesion site environment. Overexpression of col12a1a rescues the effects of Wnt/β-catenin pathway inhibition and is sufficient to accelerate regeneration. We demonstrate that in a vertebrate of high regenerative capacity, Wnt/β-catenin signaling controls the composition of the lesion site extracellular matrix and we identify Collagen XII as a promoter of axonal regeneration. These findings imply that the Wnt/β-catenin pathway and Collagen XII may be targets for extracellular matrix manipulations in non-regenerating species.

[1] Centre for Neuroregeneration, University of Edinburgh, The Chancellor's Building, 49 Little France Crescent, Edinburgh EH16 4SB, UK. [2] Institute for Immunology, TechnischeUniversität Dresden, Fetscherstraße 74, Dresden 01307, Germany. [3] Institute of Biochemistry and Molecular Biology, Ulm University, Albert-Einstein-Allee 11, Ulm 89081, Germany. [4] Technische Universität Dresden, DFG-Center of Regenerative Therapies Dresden, Cluster of Excellence at the TU Dresden, Fetscherstraße 105, Dresden 01307, Germany. Thomas Becker and Catherina G. Becker contributed equally to this work. Correspondence and requests for materials should be addressed to T.B. (email: Thomas.Becker@ed.ac.uk) or to C.G.B. (email: Catherina.Becker@ed.ac.uk)

After a spinal injury, non-neural cells invade the lesion site and set up a complex extracellular matrix (ECM)[1, 2]. The lesion site ECM consists of a plethora of molecules that may be inhibitory, permissive, or promoting for axon re-growth. In mammals, the balance of ECM molecules is shifted towards an inhibitory matrix that does not allow axons to cross the lesion site, in turn preventing recovery of function[1, 2]. In vertebrates capable of functional spinal cord regeneration, such as fishes and salamanders, the lesion site ECM naturally allows crossing of regenerating axons, offering potential insight into how this ECM is controlled and which ECM molecules may promote regeneration[3–6].

A candidate to control the composition of the lesion site ECM in zebrafish is the Wnt/β-catenin pathway, because Wnt/β-catenin signaling has been shown to control fibroblast biology and ECM deposition[7–10]. Moreover, axonal reconnection, which is necessary for functional recovery in adult[11] and larval zebrafish[12, 13], also depends on Wnt/β-catenin pathway activity in both systems[14, 15]. Wnt/β-catenin signaling involves extracellular Wnt ligand-induced stabilization and translocation of β-catenin into the nucleus where it modifies target gene transcription together with Tcf/Lef family proteins[16]. To understand how Wnt/β-catenin signaling promotes spinal cord regeneration, cell types of action and downstream mechanisms of pathway activation, such as deposition of ECM factors, need to be identified.

Different ECM molecules have specific functions for axon growth and regeneration. For example, among the collagens, *col19a1, col18a1*, and *col15a1b* promote growth of developing motor axons, whereas only *col4a5* is necessary for the correct growth of regenerating motor axons in zebrafish[17–20]. Hence, it is important to identify specific axon re-growth-promoting ECM components in the lesion site.

Adult and larval zebrafish show robust axonal and functional regeneration after complete spinal cord transection[11–13, 21]. Importantly, axon regeneration in larvae is rapid and can be observed in real-time. Lesion-induced paralysis at 3 days-post fertilization (dpf) is followed by recovery of touch-evoked swimming within 2 days post-lesion (dpl)[13]. Swimming capacity is lost upon re-transection[15, 16]. This makes zebrafish larvae a powerful tool to identify mechanisms responsible for establishing a lesion environment that is conducive to axon growth. Moreover, sensitive reporter lines and inducible cell-type specific pathway inhibition using the TetON system can be used to interrogate the Wnt/β-catenin signaling pathway in zebrafish[22–25].

Here, we find that Wnt/β-catenin signaling in fibroblast-like cells accumulating in the lesion site is pivotal for axon re-growth and functional regeneration. A gene expression screen identifies *col12a1a/b* expression and deposition of Collagen XII downstream of Wnt/β-catenin signaling to be required for regeneration. Hence, Wnt/β-catenin signaling in non-neural cells controls the composition of an axon-growth promoting ECM environment in the spinal lesion site in zebrafish.

## Results

**Axonal bridging correlates with functional recovery.** To identify molecular mechanisms regulating the spinal lesion ECM, we first analysed dynamics of axon regeneration in zebrafish larvae. Using anti-acetylated Tubulin immunohistochemistry we observed little to no axonal re-growth within the first 12 h after a spinal lesion. At 12 h post-lesion (hpl) mainly axonal debris, indicated by fragmented anti-acetylated Tubulin immunoreactivity, and only very few and thin re-growing axon fascicles were found in the lesion site (5/20 animals; Supplementary Fig. 1a). We found no animals with a completely bridged lesion site (0/20 animals). Analysis of stills from time-lapse video

microscopy imaging at this time point confirmed little axonal re-growth (2/9 animals with axons in the lesion site, 0/9 animals with axonal bridge; Supplementary Movies 1 and 2). After this initial lag phase axons rapidly re-grew (Supplementary Fig. 1a). At 1 dpl, 57% of all larvae showed axonal bridging of the lesion site, as indicated by continuous axonal labelling between rostral and caudal spinal cord ($n = 112$). Bridging further increased to >80% of the animals at 2 dpl ($n = 111$; Fig. 1c). Remarkably, we found that in the 18% of lesioned larvae in which no axonal bridge was established by 2 dpl, functional recovery, as measured by touch-evoked swim distance, was also impaired (analysed in *Xla.Tubb*:DsRED[26] transgenic fish; Supplementary Fig. 1b,c). This underscores the importance of axonal re-growth for functional recovery in larvae.

**Axons navigate a non-neural lesion site that is rich in ECM.** To determine how axons cross the lesion site, we analysed their relationship with astroglia-like processes, as these have been shown to be supportive for axon regeneration in zebrafish[27, 28]. We imaged axonal and glial processes in time-lapse video microscopy using double transgenic fish for *Xla.Tubb*:DsRED, to visualize neuronal processes, and either *gfap*:GFP[29] ($n = 6$ fish, 33 fasicles) or *her4.3*:EGFP[30] ($n = 3$ fish, 30 fasicles) to label astroglia-like processes. This revealed that the majority of fascicles that entered the lesion site were composed of axons without detectable glial processes (49–53%). Some fascicles contained both glial and axonal processes (20–21%) and some pure glial fascicles were observed (27–30%; Fig. 1a and Supplementary Movies 1 and 2; see Methods section for scoring criteria). To confirm these findings we used a range of combinations of immunohistochemical and transgenic markers for neuronal and glial processes in static images of live animals and histological preparations at 1 dpl. This showed that 63–77% of axon-containing fascicles ($\triangleq$46–62% of all fascicles analysed) were devoid of detectable glial processes (Fig. 1b). Nineteen to 26% contained both axonal and glial processes and 19–28% of all analysed fascicles were only composed of glial processes. Moreover, double-immunohistochemistry for anti-acetylated Tubulin and anti-GFAP showed that at 1 dpl, only 36% of animals had glial processes bridging the lesion site, in contrast to 57% of the fish with an axonal bridge. At 2 dpl, glial processes had caught up (Fig. 1c). This suggests that axons cross the lesion site more rapidly than glial processes. In summary, these observations support that many axons rapidly extend into the lesion site and cross it independently of detectable glial processes.

To experimentally test a role of astroglia-like processes in supporting axonal regeneration, we ablated the majority of glial cells using *gfap*:Gal4ff;*UAS*:NTR-mCherry transgenic fish that express the enzyme nitroreductase under regulatory sequences of the *gfap* gene[31]. Adding the prodrug Metronidazole (MTZ) at 1 dpf, which is converted by nitroreductase into a cytotoxic compound, almost completely eliminated transgene-expressing glial cells by 3 dpf (Fig. 1d)[31]. A dramatic depletion of astroglia-like processes in the lesion site was confirmed by anti-GFAP immunohistochemistry (Fig. 1e). Remarkably, this treatment did not reduce the proportion of animals with axonal bridge (anti-acetylated Tubulin[+]; Fig. 1e). This further supports that glial processes do not have a major role in facilitating axonal re-growth across a spinal lesion site. However, we cannot rule out that glial processes that were not detected with our markers and were spared by the ablation still supported axonal regeneration.

To explore possible substrates for the highly dynamic growth cones other than glia, we determined regulation and protein deposition for two major ECM components, *collagen1a2* (*col1a2*) and *fibronectin1a* (*fn1a*). Both genes showed robust upregulation

of expression in the lesion site (Fig. 1f). Using specific antibodies we indeed observed an accumulation of Collagen I (Col I) and Fibronectin proteins in the lesion matrix (Fig. 1g). The Col I and Fibronectin immunoreactivity appeared diffuse, but also formed fibrous structures, often with a longitudinal orientation.

Double immunohistochemistry at 1 dpl revealed close apposition of regenerating axonal fascicles (anti-acetylated Tubulin[+]) with anti-Col I (11 of 13 fascicles associated with Col I immunoreactivity; $n = 7$ fish) or anti-Fibronectin (15 of 22 fascicles associated with Fibronectin immunoreactivity; $n = 6$

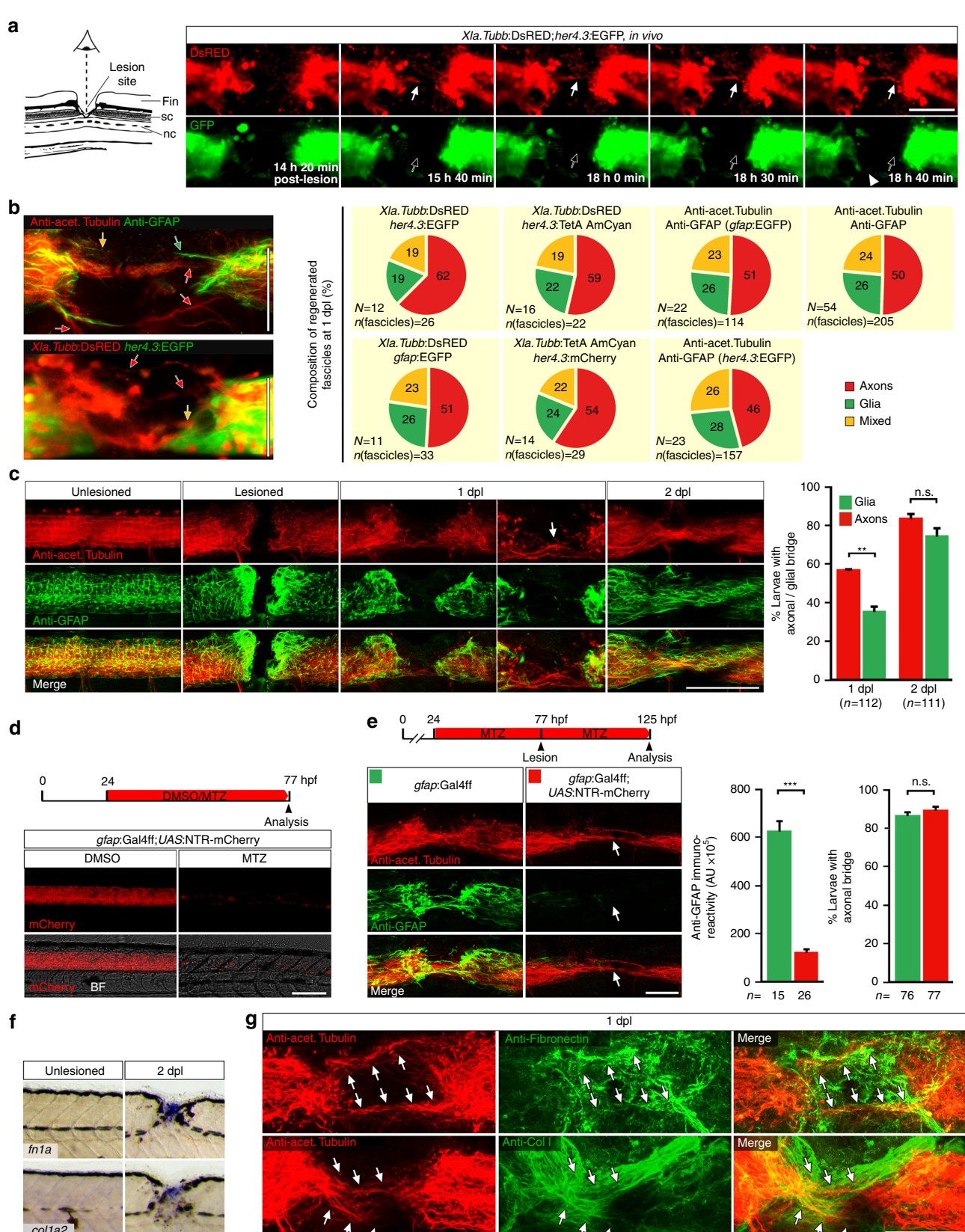

fish) immunoreactive fibres in the lesion site (arrows in Fig. 1g, Supplementary Movies 3 and 4). Hence, regenerating axons navigate the non-neural lesion environment in close association with ECM, suggesting that axon-ECM interactions might be important for axonal re-growth.

**Wnt signaling is activated after spinal cord lesion.** To determine a potential role of the Wnt/β-catenin pathway in controlling the composition of the spinal lesion ECM, we first characterized the cells in which the pathway was active after lesion. We used in situ hybridisation to detect transcripts of three different transgenic reporter lines of Wnt/β-catenin-dependent transcription, *6xTCF:dGFP, Top:dGFP, 7xTCF:mCherry*[22–24]. We also determined expression levels of direct Wnt/β-catenin target genes *axin2* and *lef1*[32, 33], which are commonly used as read-outs of pathway activation and act as feedback inhibitor (*axin2*) and transcriptional activator (*lef1*) in the pathway, respectively (see Fig. 2a). All probes showed increased labelling in the lesion site at 2 dpl (Fig. 2b; Supplementary Fig. 2a,b). Time course analysis in the highly sensitive *6xTCF:dGFP* Wnt reporter line indicated incipient pathway activity in the lesion site at 0.5 dpl (12 hpl; arrows in Supplementary Fig. 2c), increased activity at 1 dpl, maximal activation at 2 dpl and return to undetectable control levels by 5 dpl after dorsal incision lesion or lateral stab lesion (Fig. 2b, Supplementary Fig. 2d,e). This correlates well with the time course of axonal regeneration. The *6xTCF:dGFP* reporter also labelled a few additional cells within the spinal cord at 1 and 2 dpl (empty arrows in Fig. 2c). These cells have previously been described as glial cells[14].

*6xTCF:dGFP* reporter labelling was specific for Wnt/β-catenin pathway activity in our experimental paradigm, as pharmacological inhibition (IWR-1) or heat shock-induced ubiquitous overexpression of the negative regulator *axin1* in *hs:Axin1*[34] transgenic larvae strongly reduced the signal in the lesion site (Supplementary Fig. 2f,g). Importantly, using anti-GFP immunohistochemistry on whole mounts and sections confirmed the strong signal in the lesion site and only sparse labelling in the adjacent spinal cord in *6xTCF:dGFP* transgenic animals (Supplementary Fig. 2h,i). Moreover, our in situ hybridization protocol allowed us to detect *gfp*-expressing glial cells throughout the spinal cord in whole mount preparations of 5 dpf *her4.3*:EGFP transgenic fish (Supplementary Fig. 2j). This shows that lack of labelling in neural tissue was not due to potential probe penetration issues in whole mount preparations. Hence, after spinal cord lesion, Wnt/β-catenin signaling is mainly increased in the non-neural lesion site, which is a major, previously undescribed, domain of Wnt/β-catenin pathway activity.

**Wnt signaling is active mainly in fibroblast-like cells.** To characterize the Wnt-responding cell types in the non-neural lesion site environment, we utilized *6xTCF:dGFP* transgenic fish. GFP protein and mRNA was mainly detectable at the lesion edge at 1 and 2 dpl (Fig. 2c; Supplementary Fig. 2h). Wnt-responding cells did not co-label with markers of muscle fibres (MyHC/F59[35], n > 10; Supplementary Fig. 3a,b) or immune cells (L-Plastin[36], n > 10; Supplementary Fig. 3c). In contrast, 31% of GFP-labeled cells (cell counts in immunolabeled sections of individual animals, n = 10) were co-labeled with the basal keratinocyte marker p63[37] (arrowheads in Fig. 2d; Supplementary Fig. 3d). The remaining two thirds of the Wnt-responsive cells (*6xTCF:dGFP+/p63−*) were located subepidermally in the lesion site and could thus be fibroblasts. There is no definitive fibroblast marker. However, fibroblasts are known to express the ECM core components *col1a2* and *fn1a* during dermal stroma formation in developing zebrafish[38, 39]. Indeed we found that at 1 dpl and 2 dpl, Wnt-responding subepidermal cells were positive for *col1a2* and *fn1a* transcripts (arrows in Fig. 2e; Supplementary Fig. 3e–g), which were both upregulated in the lesion site (Fig. 1f). Interestingly, in the peripheral lesion area, where the myotome was still intact, a population of *6xTCF:dGFP+/p63−* cells, continuous with the subepidermal cells in the lesion site, were found sandwiched between basal keratinocytes (p63+) and slow twitch muscle fibres (MyHC/F59+) (insets in Supplementary Fig. 3a,b). Additional *6xTCF:dGFP+/p63−* cells were found covering the spinal cord (arrows in Supplementary Fig. 3h). These positions correspond to previously described dermal and meningeal fibroblast locations in the developing zebrafish and adult eel[39, 40]. Based on these observations, we tentatively identify the vast majority of Wnt-responding cells as fibroblast-like. Importantly, we found fibroblast-like Wnt-responding cells (*6xTCF:dGFP+/p63−*) in close proximity to regenerating axons (anti-acetylated Tubulin+) in the spinal lesion site at 1 dpl (arrows in Fig. 2f). Hence, the majority of Wnt-responding cells after a lesion are non-neural fibroblast-like cells that are part of the lesion environment through which axons navigate.

**Regeneration requires Wnt signaling in non-neural cells.** To functionally inhibit Wnt/β-catenin pathway activity selectively in lesion site cells, we took advantage of the TetON system that allows for Doxycycline (DOX)-inducible (Supplementary Fig. 4a), cell type-specific pathway manipulation[41]. To interfere with Wnt/β-catenin signaling, we used *TetRE:Axin1-YFP* transgenic fish, a TetResponder line that has been shown to efficiently inhibit the pathway when activated during zebrafish development or adult tail fin regeneration[25, 41]. To specifically target cells that activate

**Fig. 1** Regenerating axons navigate a non-neural lesion environment that is rich in ECM independently of astroglia-like processes. **a** Time-lapse video-microscopy reveals that axonal growth cones (arrow, *Xla.Tubb*:DsRED) extend into the lesion site independently of astroglia-like processes (*her4.3*:EGFP, *empty arrow* points out lack of glial labelling). *Arrowhead* indicates glial process extending into the lesion site. Single frames are shown. Abbreviations: fin, dorsal fin fold; sc, spinal cord; nc, notochord. **b** Quantification of fascicle composition at 1 dpl using different combinations of immunohistochemical and transgenic markers for astroglia-like processes and axons indicates that the majority of fascicles are neuronal with no detectable glial component (46–62% of all fascicles analyzed). In the example scans on the left, *yellow arrows* indicate mixed fascicles, containing neurites and glial processes; *green arrows* indicate pure glial fascicles and *red arrows* indicate pure axonal fascicles. **c** Time course of re-growth of axons (anti-acetylated Tubulin+) and astroglia-like processes (anti-GFAP+) after spinal cord transection. Quantification of labelling continuity between rostral and caudal spinal cord stumps at 1 dpl and 2 dpl suggests faster bridging of the lesion site by neuronal rather than glial processes. *Arrow points* to axonal bridge with very few glial processes in the lesion site at 1 dpl at higher magnification (Fischer's exact test: **P < 0.01; n.s. indicates not significant). **d** Treatment of *gfap*:Gal4ff;*UAS*:NTR-mCherry transgenic fish with the pro-drug Metronidozole (MTZ) almost completely ablates transgene-expressing glial cells by 3 dpf. **e** Treatment of *gfap*:Gal4ff;*UAS*:NTR-mCherry transgenic fish with MTZ dramatically reduces anti-GFAP+astroglia-like processes (*t*-test: ***P < 0.001) but does not reduce the proportion of larvae with axonal bridges at 2 dpl (Fischer's exact test: n.s indicates not significant). **f** *fn1a* and *col1a2* expression is upregulated after lesion, shown by in situ hybridization. **g** Regenerating axons (anti-acetylated Tubulin+; *arrows*) are closely associated with Collagen I and Fibronectin immunoreactivity in a lesion site (confocal depth was limited to spinal cord). Also see Supplementary Movies 3 and 4. **a–g** Views are dorsal (**a**; rostral is *left*) or lateral (b-g; dorsal is *up*, rostral is *left*). BF: brightfield. *Scale bars*: 100 μm **a,c–f** and 50 μm **b,g**. *Error bars* indicate s.e.m

the Wnt/β-catenin pathway in the lesion site, we generated a TetActivator line using regulatory elements of the direct Wnt/β-catenin target *lef1* (*lef1*:TetA AmCyan; Supplementary Fig. 4b). Indeed, this strategy showed injury-induced TetActivator expression (*amcyan* mRNA) in Wnt-responding cells in the lesion site in *lef1*:TetA AmCyan;*6xTCF*:dGFP double-transgenic fish (Supplementary Fig. 4c). YFP fluorescence was likewise concentrated in the lesion site in *lef1*:TetA AmCyan;*TetRE*:Axin1-

YFP double-transgenic fish after DOX treatment (Fig. 3a,b). This indicates successful selective targeting of Wnt-responding cells in the lesion site. The inhibition was effective, as lesion site expression of *axin2*, an established read-out for Wnt/β-catenin pathway activity[33] (Fig. 2a), was strongly reduced (Supplementary Fig. 4d).

To determine whether functional spinal cord regeneration was impaired in *lef1*:TetA AmCyan; *TetRE*:Axin1-YFP

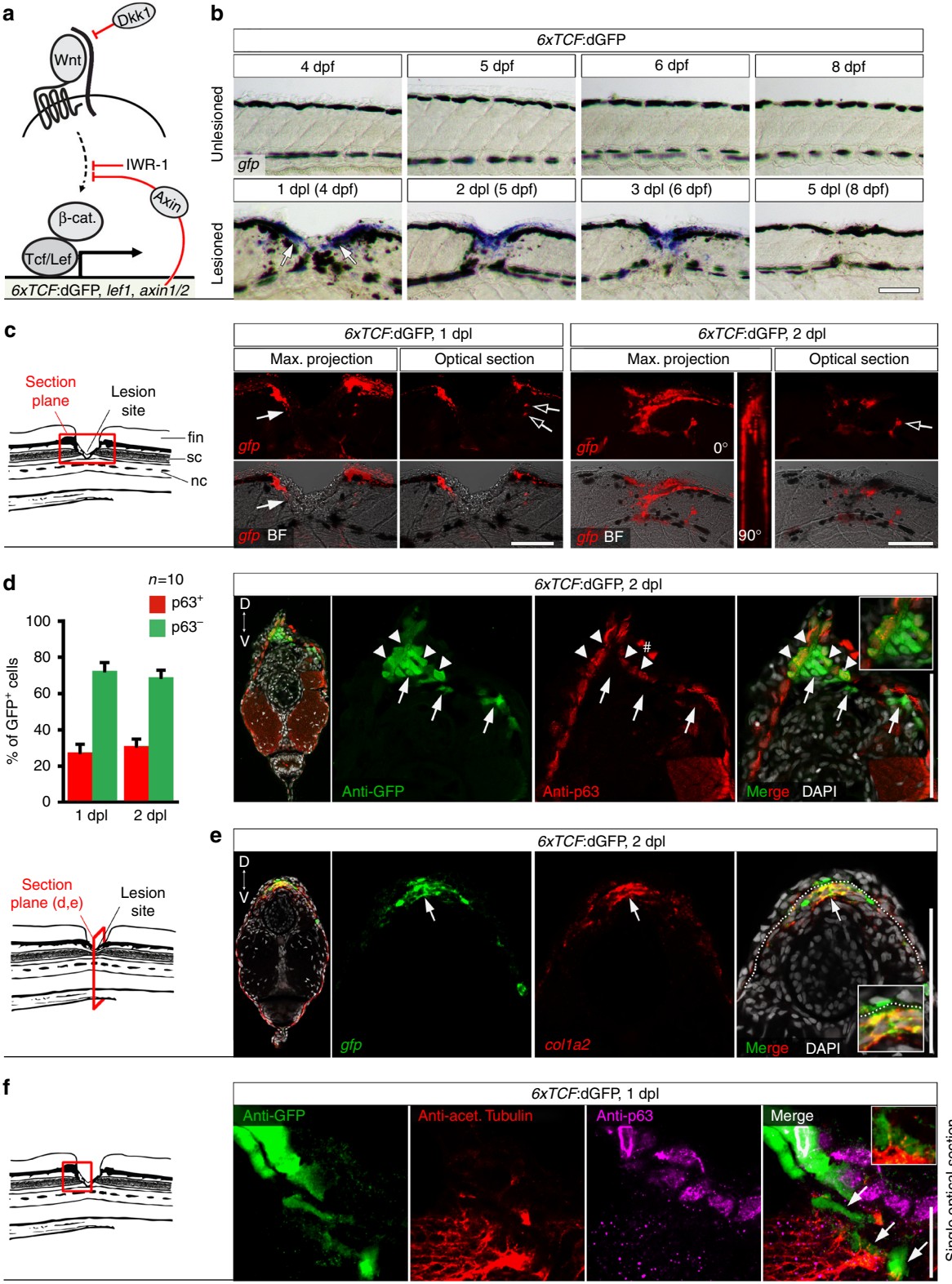

double-transgenic fish after DOX treatment, we scored fish showing continuity of axonal labeling over the lesion site (anti-acetylated Tubulin[+]) and quantified swim distance after touch at 2 dpl. The proportion of larvae with bridged lesion sites was reduced by 48% (Fig. 3c). Moreover, larvae recovered to swim only less than half the distance (42%) of unlesioned Wnt-inhibited animals, whereas lesioned control larvae recovered to swim the same distance as their unlesioned age-matched controls (Fig. 3d).

Importantly, neuronal (*Xla.Tubb*:TetA AmCyan) and glial (*her4.3*:TetA AmCyan[25])-specific Axin1 induction did not induce a phenotype, whereas ubiquitous expression (*ubi*:TetA AmCyan[25]) recapitulated the axonal and behaviour phenotypes of lesion site-specific manipulations (Fig. 3e–m; Supplementary Fig. 4e). Of note, Axin1 was correctly targeted in all of these lines, as indicated by YFP fluorescence of the Axin1-YFP fusion protein in neurons (*Xla.Tubb*:TetA AmCyan), glia (*her4.3*:TetA AmCyan) or throughout the larvae (*ubi*:TetA AmCyan). Moreover, *axin1* induction in neural cells reached functional levels. This is indicated in the neuronal-driven TetActivator line (*Xla.Tubb*:TetA AmCyan), in which *axin2* expression in the lesion site remained unaffected but was strongly reduced in a constitutively Wnt-responsive domain in the brain (Supplementary Fig. 4f,g). Hence, Wnt/β-catenin pathway activity in neurons or glia is not detectably involved in axonal regeneration. To exclude non-specific effects of the TetON system or Axin1-YFP induction, we also interfered systemically with Wnt/β-catenin signaling using a pharmacological approach (IWR-1) or via heat shock-induced ubiquitous overexpression of the pathway antagonists *axin1* or *dkk1* in *hs*:Axin1 or *hs*:dkk1[42] transgenic fish, respectively. Using *dkk1* overexpression as an additional control was important, as it specifically antagonizes the pathway at the receptor level, whereas *axin1* overexpression and IWR-1 promote degradation of the transcriptional co-activator β-catenin (Fig. 2a; Supplementary Note 1). All of these manipulations robustly reduced the proportion of larvae with an axonal bridge (reduction by 51–72%) and functional recovery (≤ 48% of the swim distance covered by unlesioned Wnt-inhibited animals; Supplementary Fig. 5a–c). Inhibition of Wnt/β-catenin signaling (IWR-1 treatment) starting at different time points post-lesion showed that axon re-growth was still maximally inhibited when the treatment was started at 0.5 dpl (12 hpl), but the effect of IWR-1 was greatly diminished when the treatment was started at 1 dpl and had no effect when it was started at 1.5 dpl (Supplementary Fig. 5d). Hence, Wnt/β-catenin pathway activity is required mainly between 0.5 dpl and 1 dpl for axon regeneration, when axons actively grow across the lesion site.

Two lines of evidence support that effects of Wnt/β-catenin pathway inhibition on axon regeneration were not merely a secondary consequence of potential defects in wound healing. Firstly, a lateral stab lesion, which generates minimal tissue disruption (Supplementary Fig. 2e), showed a similar reduction in axonal regeneration and functional recovery after pharmacological Wnt/β-catenin pathway inhibition, as was observed after dorsal incision lesion (Supplementary Fig. 5e). Second, wound re-epithelialization by basal keratinocytes, providing a protective layer over the lesion site[43], robustly occurred in controls (30/30) and in larvae in which Wnt/β-catenin signaling had been systemically inhibited using IWR-1 (30/30) (visualized in *krtt1c19e*:EGFP transgenic fish[44]; arrow in Supplementary Fig. 6; also see Fig. 4h). We conclude that Wnt/β-catenin signaling in fibroblast-like cells in the lesion site, but not in neural cells, is necessary for functional axon regeneration.

Interestingly, systemic but not glial cell-specific Wnt-inhibition also reduced the number of animals with glial bridges. This suggests that glial crossing of the lesion site also depends on Wnt/β-catenin signaling in a non-cell autonomous way (Supplementary Fig. 7a,b).

**Wnt signaling promotes Col XII deposition**. We hypothesized that Wnt/β-catenin signaling in lesion site fibroblast-like cells controls ECM deposition, as fibroblasts are a major source of ECM under physiological and pathological conditions. To identify relevant Wnt/β-catenin-controlled ECM components, we compared labelling intensities between control and IWR-1-treated stab-lesioned larvae in an in situ hybridisation screen for two fibronectins and 44 collagen chain-encoding genes at 1 dpl (Fig. 4a; Supplementary Fig. 8a). These genes were chosen, because of roles of these ECM protein classes in axon growth and regeneration[17–20, 45]. In DMSO-treated controls we found expression of both fibronectins and 10 collagens to be locally upregulated in the lesion site compared to adjacent unlesioned trunk tissue. Of these upregulated genes, only expression of *col6a2* and of the paralogs *col12a1a* and *col12a1b* were noticeably reduced when Wnt/β-catenin signaling was inhibited (IWR-1 treatment). We did not find any increased expression of ECM molecules after IWR-1 treatment. We focused our further analysis on *col12a1a* and *col12a1b*, because both paralogs code for the essential α-chain of Collagen XII (Col XII) and are regulated by Wnt/β-catenin signaling. Since functional Col XII molecules assemble as α-chain trimers, Col XII protein expression likely also depends on Wnt/β-catenin signaling, as no other paralogs were found in the genome that could substitute for *col12a1a* and *col12a1b*[46].

First we determined the time course of *col12a1a/b* expression after a lesion. Parallel to the time course of Wnt/β-catenin pathway activation, *col12a1a/b* transcripts were detectable in the lesion site at 0.5 dpl (12 hpl). Further increased expression was detected at 1 dpl, when axons started to regenerate. Expression peaked at 2 dpl and had returned to baseline levels by 5 dpl (Fig. 4b; Supplementary Fig. 8b).

To verify control of *col12a1a/b* expression by Wnt/β-catenin signaling, we quantified fluorescence in situ hybridisation signals for *col12a1a/b* transcripts at 1 dpl after 24 h of IWR-1 treatment. This confirmed transcriptional downregulation of *col12a1a* and

**Fig. 2** Wnt/β-catenin signaling is mainly active in fibroblast-like cells after spinal cord lesion. **a** Cartoon showing relevant transcriptional read-outs of Wnt/β-catenin pathway activity, feedback regulatory mechanisms and strategies to interfere with signaling at different levels of the pathway. Abbreviation: β-cat., β-catenin. **b** Detection of *gfp* mRNA in the *6xTCF*:dGFP transgenic reporter line shows transient activity of the Wnt/β-catenin pathway in the lesion site during regeneration. **c** *gfp* expression is largely confined to the lesion edge in *6xTCF*:dGFP transgenic larvae (*arrow*). A few additional cells are labelled within the presumptive spinal cord region (*empty arrows*). Abbreviations: nc, notochord; sc, spinal cord; fin, dorsal fin fold. **d** The *6xTCF*:dGFP Wnt reporter is mostly active in subepidermal p63[−] cells (*arrows*; ≥ 69% of all *6xTCF*:dGFP[+] cells at 1 dpl and 2 dpl), but also in p63[+] basal keratinocytes (*arrowheads*) in the lesion site. [#] indicates non-specific signal. **e** Consistent with a fibroblast identity, subepidermal cells (below *dashed line*) co-express *col1a2* (*red*) and *gfp* mRNA (*green*) in a lesioned *6xTCF*:dGFP transgenic animal (*arrow*). *Dashed line* indicates junction between basal epithelium and adjacent connective tissue. **f** Fibroblast-like (*6xTCF*:dGFP[+]/p63[−]) cells are found in close proximity to regenerating axons (anti-acetylated Tubulin[+]) in the spinal lesion site at 1 dpl (*arrows*). A single optical section is shown. **a–f** Views are lateral (**b,c,f**; dorsal is *up*, rostral is *left*) or transversal (**d,e**; dorsal is *up*) as indicated. BF: brightfield. *Scale bars*: whole mounts, 100 μm **b,c** and 25 μm **f**; sections: 100 μm. *Error bars* indicate s.e.m

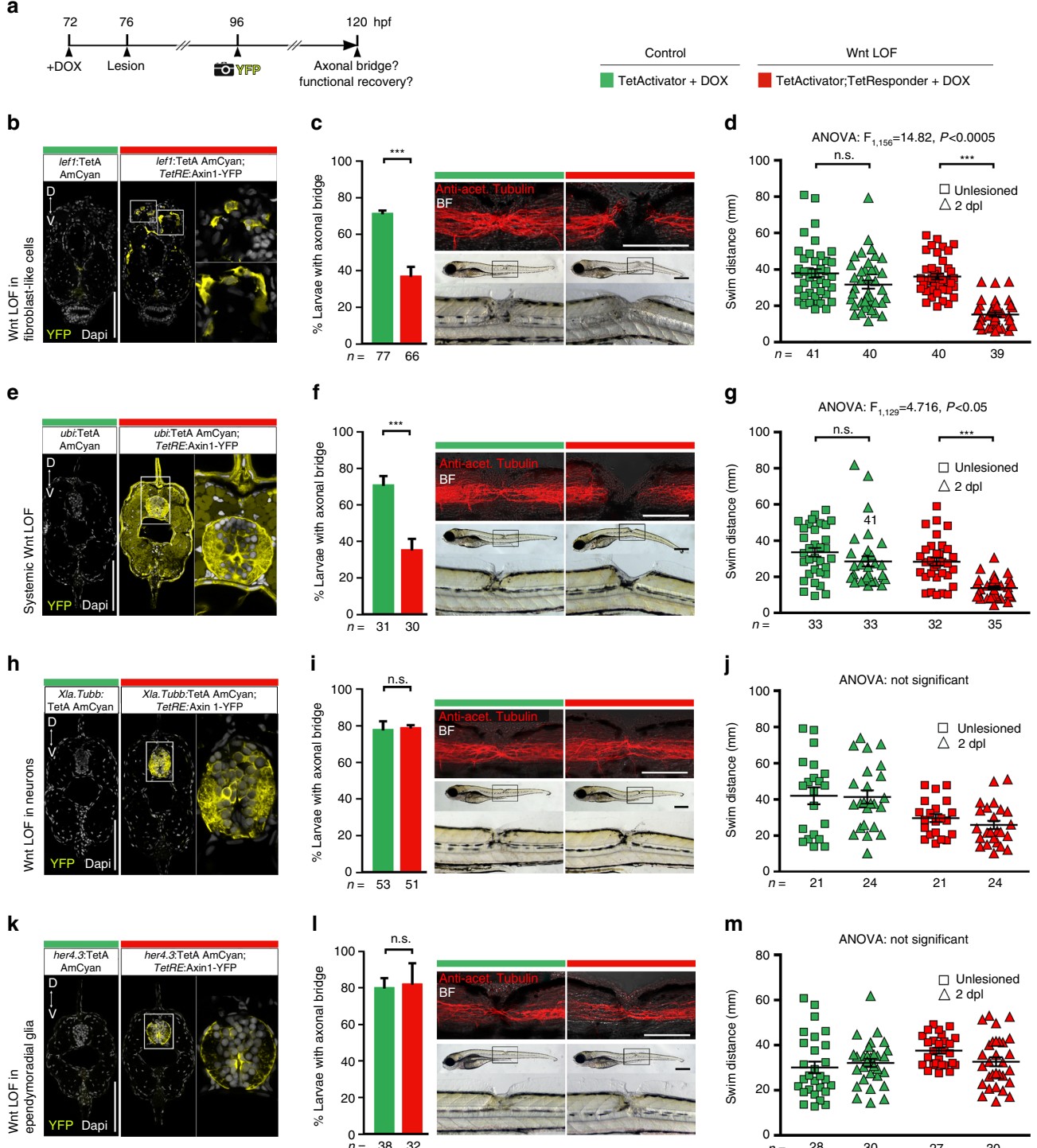

**Fig. 3** Wnt/β-catenin signaling in non-neural cells is required for axon regeneration and functional recovery. **a** Schematic timeline of the experimental design using the TetON system. **b–d** *axin1* overexpression selectively in cells in the lesion site (**b**; *lef1*$^+$) is sufficient to interfere with axon regeneration (**c**; Fischer's exact test: ***$P < 0.001$) and functional recovery (**d**; two-way ANOVA: $F_{1,156} = 14.82$, $P < 0.0005$; *t*-test: ***$P < 0.001$). **e–g** *axin1* overexpression throughout the fish **e** interferes with axon regeneration (**f**; Fischer's exact test: *$P < 0.05$) and functional recovery (**g**; two-way ANOVA: $F_{1,129} = 4.716$, $P < 0.05$; *t*-test: ***$P < 0.001$). **h–j** Neuronal *axin1* overexpression, indicated by YFP labelling of parenchymal cell bodies and neuropil in the spinal cord **h**, does not interfere with axon regeneration **i** or functional recovery (**j**; two-way ANOVA shows no interaction between lesion and treatment). **k–m** *axin1* overexpression in ependymoradial glial cells, indicated by YFP labeling in cell bodies lining the central canal, their radial processes and pial end feet **k** does not interfere with axon regeneration **l** or functional recovery (**m**; two-way ANOVA shows no interaction between lesion and treatment). **b–e** Views are transversal (**b,e,h,k**; dorsal is *up*) or lateral (**c,f,i,l**; dorsal is *up*, rostral is *left*). BF: brightfield. n.s.: not significant. *Scale bars*: whole mounts, 200 and 100 μm; sections: 100 μm. *Error bars* indicate s.e.m

*col12a1b* by 88% and 98% respectively (Fig. 4c). RT-PCR of lesion site-enriched trunk tissue confirmed that IWR-1 treatment interfered with lesion-induced *col12a1a* and *col12a1b* upregulation at 1 dpl, while *col1a1a* expression levels remained unaffected by Wnt/β-catenin pathway inhibition (Supplementary Fig. 8c). Consistent with our finding that Wnt/β-catenin pathway activity

is mainly required within the first 24 h after a lesion for axonal re-growth (Supplementary Fig. 5d), IWR-1 treatment for 24 h between 1 dpl and 2 dpl had no effect on *col12a1a/b* expression. (Supplementary Fig. 8d). This indicates that sustained Wnt/β-catenin pathway activity was not needed for *col12a1a/b* transcription.

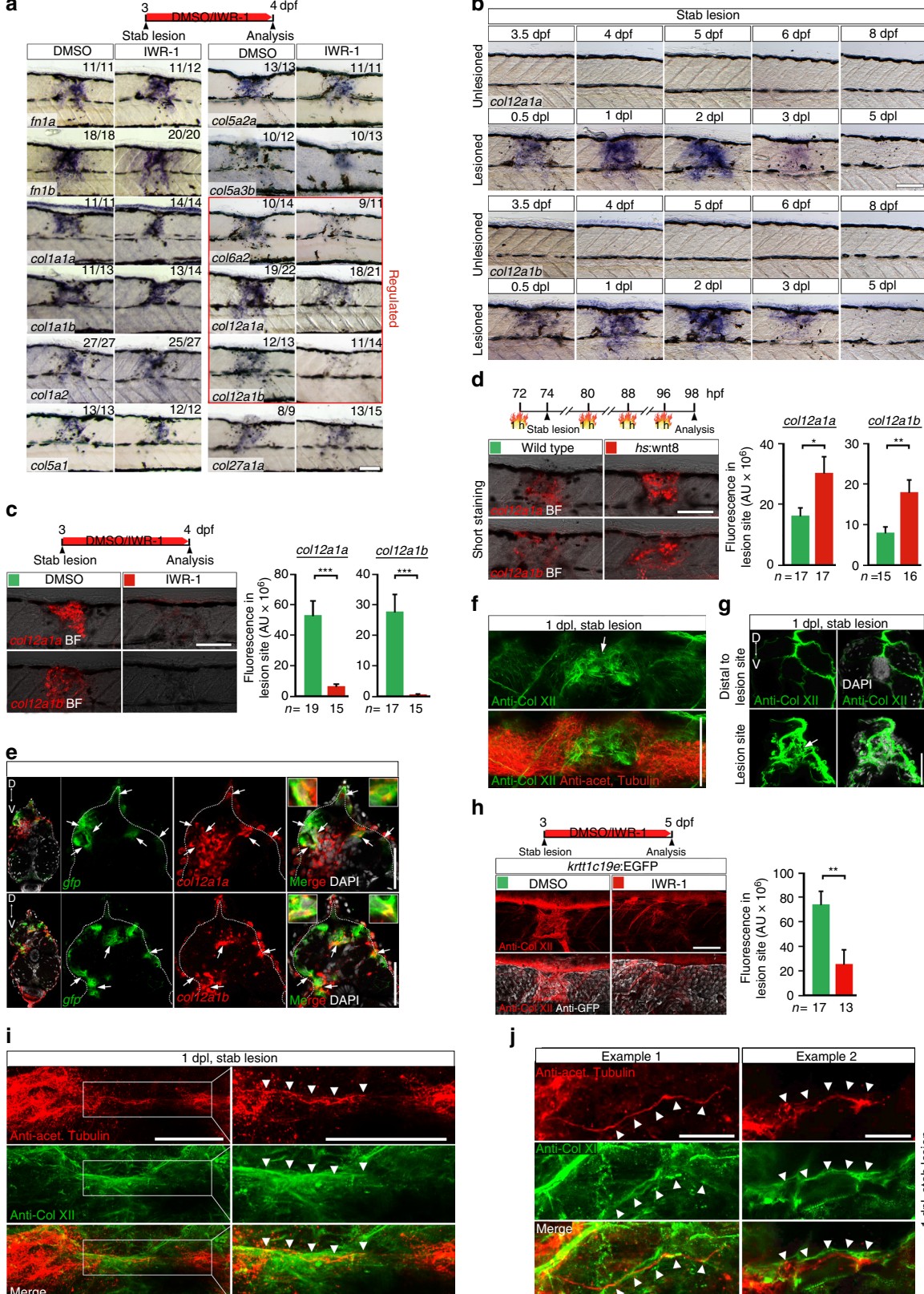

Activation of Wnt/β-catenin signaling through overexpression of *wnt8* for 24 h in *hs*:wnt8 transgenic fish[47] enhanced expression of both *col12a1* genes at 1 dpl specifically in the lesion site (Fig. 4d). Thus, gain- and loss-of-function experiments for Wnt/β-catenin pathway activity support that zebrafish orthologs of the *col12a1* gene are transcriptionally controlled by Wnt/β-catenin signaling in a spinal lesion site in zebrafish larvae during the initial phase of axon regeneration.

To determine whether Wnt/β-catenin signaling regulates *col12a1a/b* transcription in the lesion site likely directly or indirectly, we compared the kinetics of down-regulation of *col12a1a/b* expression after pathway inhibition with that of the direct transcriptional target gene *axin2* at 1 dpl. Short-term inhibition of Wnt/β-catenin pathway activity through *axin1* overexpression in *hs*:Axin1 transgenic fish for 6 h was sufficient to abolish expression of *axin2* while *col12a1a* and *col12a1b* expression was not affected at that early time point (Supplementary Fig. 8e). This indicates delayed and thus indirect control of *col12a1a/b* transcription by Wnt/β-catenin signaling. Taken together, these data support a scenario in which Wnt/β-catenin signaling promotes axonal regeneration by initiating *col12a1a* and *col12a1b* expression after a lesion.

To characterize the *col12a1a* and *col12a1b*-expressing cells in the lesion site, we used multicolour fluorescence in situ hybridization. The majority of *col12a1a* and *col12a1b*-expressing cells co-labeled with *pdgfrb*:GFP[48] reporter transcripts, a marker for pericytes and reactive fibroblasts, at 1 dpl (Supplementary Fig. 8f,g)[49]. Inhibition of Wnt/β-catenin signaling (IWR-1 treatment) did not overtly inhibit appearance of *pdgfrb*:GFP+ cells, indicating that Wnt/β-catenin signaling does not act on potential migration of *col12a1a/b*-expressing cells into the lesion site (Supplementary Fig. 8h). Some of the *col12a1a* and *col12a1b*-expressing cells were also positive for *6xTCF*:dGFP reporter transcripts at 1 dpl (arrows in Fig. 4e). However, a substantial proportion of *col12a1a/b*-expressing cells were not co-labelled with *6xTCF*:dGFP reporter transcripts, indicating that expression of *col12a1a* and *col12a1b* was not exclusive to acutely Wnt-responding cells. Hence, Wnt/β-catenin pathway activity could have already ceased in the majority of *col12a1a* and *col12a1b*-expressing cells at the time point of analysis or Wnt-responding cells could induce additional non-neural cells in the lesion site to express these genes.

We next assessed whether Col XII protein deposition was altered by inhibition of Wnt/β-catenin signaling. We found increased immunoreactivity of Col XII in the lesion site at 1 dpl and 2 dpl in control lesioned larvae compared to adjacent unlesioned trunk tissue or unlesioned control animals (arrows in Figs. 4f,g and Supplementary Fig. 9a). Inhibition of Wnt/β-catenin signaling by IWR-1 treatment or after heat shock-induced

overexpression of *dkk1* in *hs*:dkk1 transgenic fish, led to lower Col XII immunoreactivity in lesioned larvae than in DMSO-treated controls (Fig. 4h, Supplementary Fig. 9b). Hence, strongly reduced levels of Col XII matrix are laid down in the lesion site under conditions of Wnt/β-catenin pathway inhibition.

To analyse the relationship of regenerating axons and Col XII, we used double-immunolabelling of axons (anti-acetylated Tubulin+) and Col XII. This showed little to no Col XII deposition in the lesion site at 0.5 dpl (12 hpl) and no axons that had entered the lesion site at this time point (asterisk in Supplementary Fig. 9c). In contrast, at 1 dpl we detected prominent anti-Col XII immunoreactivity (Supplementary Fig. 9c) and regenerating axons were present in the lesion site in close proximity to Col XII immunoreactivity (arrowheads in Fig. 4i, Supplementary Movie 5). The trajectory of axonal fascicles frequently appeared to follow longitudinal fibres of Col XII immunoreactive ECM material (arrowheads in Fig. 4j). We found 31 axonal fascicles of 45 in close apposition with Col XII positive fibres at 1 dpl ($n = 14$ fish). Thus, Col XII deposition correlates well with axonal re-growth in larvae.

In adult zebrafish, lesion site-specific expression of *col12a1a* as well as Col XII deposition was also observed at 7 dpl and 14 dpl (Supplementary Fig. 10a,b). At 14 dpl, anti-acetylated Tubulin+ spinal axons entered the lesion site in close contact with Col XII immunoreactivity, resembling the larval pattern (Supplementary Fig. 10b). Thus, after spinal cord injury, deposition of Col XII matrix in the lesion site is conserved across developmental stages in zebrafish. Taken together, this supports a scenario in which Wnt/β-catenin signaling is necessary for the correct expression and deposition of Col XII in the spinal lesion site, which in turn promotes axon regeneration.

**Col XII promotes functional spinal cord regeneration**. To determine whether Col XII deposition is required for functional axonal regeneration, we co-injected Vivo-Morpholinos (MO), which act as cell-permeable translation blocking anti-sense molecules for *col12a1a* and *col12a1b*, into the circulation for efficient distribution in 3 dpf larvae (Supplementary Fig. 11a[50]). This approach effectively reduced Col XII immunoreactivity in the lesion site compared to control MO-injected larvae (Fig. 5a,b). In contrast, immunoreactivity for Fibronectin and Collagen I was unaffected (Supplementary Fig. 11b), indicating specificity of the manipulation for *col12a1a/b* expression.

The proportion of larvae with an axonal bridge (determined in *Xla.Tubb*:DsRED transgenic animals) was reduced by 43% in *col12a1a/b* morphants compared to control MO-injected larvae at 2 dpl (Fig. 5c). In lesioned *col12a1a/b* morphants swimming

**Fig. 4** Wnt/β-catenin signaling controls *col12a1a/b* transcription and Col XII deposition in a spinal lesion site. **a** Wnt/β-catenin pathway inhibition (IWR-1) results in lower expression of select collagen genes (*red box*) in the lesion site at 1 dpl, determined by in situ hybridization. **b** *col12a1a* and *col12a1b* expression is transiently upregulated in the lesion site during regeneration. **c** Inhibition of Wnt/β-catenin signaling (IWR-1) interferes with *col12a1a/b* expression in the lesion site ($t$-test: \*\*\*$P < 0.001$). **d** *wnt8* overexpression upregulates *col12a1a/b* expression in the lesion site in *hs*:wnt8 transgenic fish ($t$-test: \*$P < 0.05$, \*\*$P < 0.01$). Signal was underdeveloped to optimize detection of differences in signal strength. **e** A subpopulation of *col12a1a* or *col12a1b*-expressing cells (*red*) in the lesion site co-labels with transcripts of the *6xTCF*:dGFP Wnt reporter (*green, arrows*). **f,g** Anti-Col XII immunoreactivity is increased in the lesion site (*arrows*) compared to adjacent unlesioned trunk tissue at 1 dpl. Shown is a confocal image of a whole mount (**f**, confocal depth was limited to spinal cord) and a transversal section though the lesion site **g**,**h** Inhibition of Wnt/β-catenin signaling (IWR-1) interferes with Col XII deposition in the lesion site ($t$-test: \*\*$P < 0.01$). Note continuity of tissue in IWR-1-treated animals as shown by re-epithelialization of the lesion site by basal keratinocytes in *krtt1c19e*:EGFP transgenic fish. **i,j** Double immunolabelling of axons (anti-acetylated Tubulin+) and Col XII shows Col XII matrix deposition in the lesion site through which regenerating axons grow **i**. Shown is a maximum intensity projection in **i** with *inset* higher magnification; the confocal depth was limited to spinal cord. *Arrowheads* point to regenerating axonal fascicle in close apposition to Col XII immunoreactivity (also see Supplementary Movie 5). Note, that the trajectory of axonal fascicles appear to follow longitudinal fibres of Col XII immunoreactive ECM material in the lesion site (**j**, *arrowheads*). **a–j** Views are lateral (**a–d,f,h–j**; dorsal is *up*, rostral is *left*) or transversal (**e,g**; dorsal is *up*). BF: brightfield. *Scale bars*: 100 μm **a–d,f,h**, 50 μm **e,g,i**, 25 μm **j**. *Error bars* indicate s.e.m

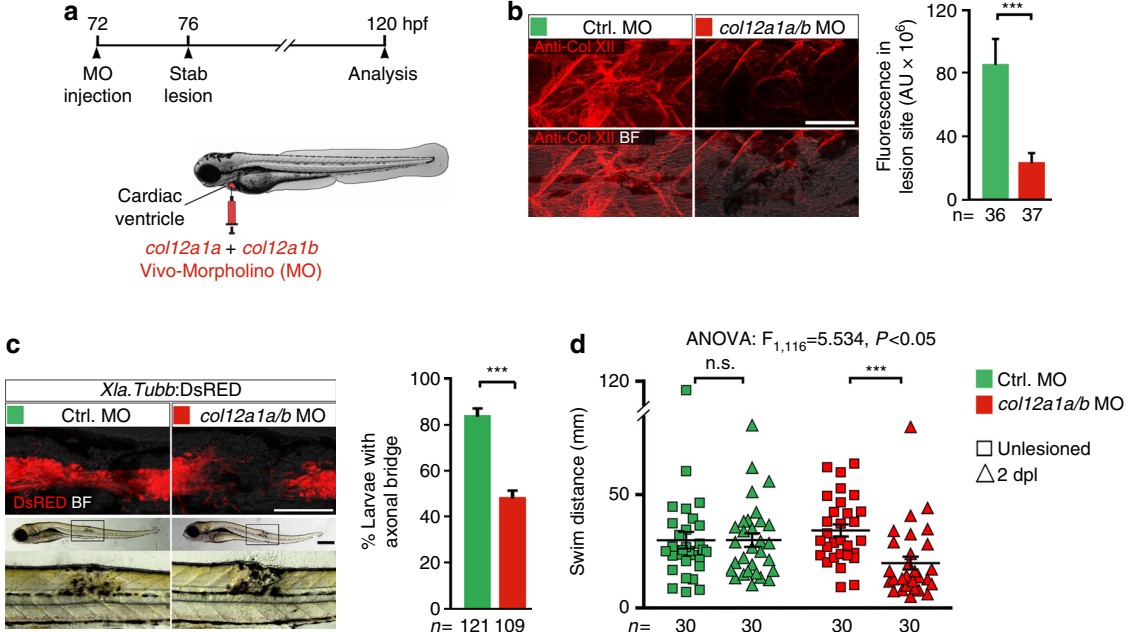

**Fig. 5** Knockdown of *col12a1a/b* inhibits functional axon regeneration. **a** Cartoon showing strategy used for systemic delivery of Vivo-Morpholino (MO), which acts as cell-permeable translation blocking anti-sense molecules. MOs against *col12a1a* and *col12a1b* were co-injected into the cardiac ventricle at 3 dpf, 4 h before lesion. **b** *col12a1a/b* MO injection strongly reduces Col XII immunoreactivity in the lesion site compared to control MO injected animals. (t-test: ***$P < 0.001$). **c** *col12a1a/b* MO injection inhibits axon regeneration after lesion (Fischer's exact test: ***$P < 0.001$). **d** *col12a1a/b* MO injection interferes with functional recovery after lesion (two-way ANOVA: $F_{1,116} = 5.534$, $p < 0.05$; t-test: ***$P < 0.001$). **a**–**d** Lateral views are shown (dorsal is *up*, rostral is *left*). BF: brightfield. n.s.: not significant. *Scale bars*: 200 μm **c** and 100 μm (**b**,**c**). *Error bars* indicate s.e.m

distance was reduced by 58% compared to unlesioned *col12a1a/b* MO-injected larvae (Fig. 5d). Importantly, Vivo-Morpholino treatment was not toxic, since lesioned control MO-injected larvae recovered to the same swimming capacity as unlesioned age-matched control MO-injected larvae at 2 dpl, and *col12a1a/b* MO treatment did not impair swimming in unlesioned animals. Impaired functional regeneration after MO knockdown supports that Col XII is part of the axon-growth promoting ECM in the lesion site.

To independently confirm the function of *col12a1a/b*, we used transient CRISPR/Cas9-mediated gene editing. Injection of small guide-RNA (gRNA) and *cas9* mRNA into the zygote efficiently generates biallelic mutations of targeted genes in a very high proportion of cells, leading to loss-of-function phenotypes[51]. Indeed, targeting of *col12a1a* and *coll12a1b* efficiently disrupted both genes in CRISPR-injected embryos, as shown by loss of targeted endonuclease restriction sites (Fig. 6a, b). Quantitative RT-PCR indicated reduced abundance of transcripts at 1 dpf, likely caused by non-sense mediated RNA decay in CRISPR-manipulated larvae (hereafter referred to as Crispants; Fig. 6c). Similarly, after a spinal lesion in Crispants the abundance of both *col12a1a* and *col12a1b* mRNAs in the lesion site was also reduced, as determined by quantification of fluorescence in situ hybridisation signals (Fig. 6d). Targeting of only *col12a1a* or *col12a1b* did not affect axonal regeneration, possibly due to compensation of the non-targeted paralog (Supplementary Fig. 12a). In contrast, in Crispants in which both *col12a1* genes were targeted, the proportion of larvae with axonal continuity (anti-acetylated Tubulin[+]) between spinal cord ends was reduced by 31% compared to controls (Fig. 6e). Single larvae analysis showed that lack of axonal bridging in *col12a1a* and *col12a1b*-targeted animals (analysed in *Xla.Tubb*:DsRED transgenic fish) correlated with high gene editing efficiency of both *col12a1* paralogs, supporting specificity of the CRISPR/Cas9 manipulation (Supplementary Fig. 12b). Hence, both MO

knockdown and CRISPR manipulation indicate an important contribution of Col XII to functional spinal cord regeneration in zebrafish.

To determine the relevance of Col XII as a downstream effector of Wnt/β-catenin signaling, we asked whether experimentally increasing expression of *col12a1a* was sufficient to rescue the regeneration phenotype induced by inhibition of the Wnt/β-catenin pathway. To this end, we created zebrafish that are transgenic for heat shock-inducible expression of full-length *col12a1a* (hs:col12a1a-p2a-Cerulean, short hs:col12a1a; Supplementary Fig. 13a). Activation of the transgene in unlesioned animals increased anti-Col XII immunoreactivity in developmental Col XII domains but also in ectopic sites, including the spinal cord (arrow in Supplementary Fig. 13b). To increase Col XII deposition in the lesion site, we applied repetitive heat shocks to hs:col12a1a transgenic animals until immediately before the lesion (3 dpf). This treatment increased detectability of transgene mRNA for at least 24 h after the last heat shock (Fig. 7a) and was sufficient to increase Col XII immunoreactivity in the lesion site of larvae in which the Wnt/β-catenin pathway was inhibited (IWR-1 treatment; analyzed at 2 dpl; arrow in Fig. 7b). We found that under these experimental conditions, *col12a1a* overexpression more than doubled the proportion of IWR-1-treated larvae with axonal bridge to 77% of all lesioned larvae, compared to 32% in only IWR-1-treated lesioned larvae (determined in *Xla.Tubb*:DsRED transgenic animals; Fig. 7c). This represents a rescue of the axonal bridging phenotype to near wild type levels (compare Supplementary Fig. 1b). *col12a1a* overexpression also improved functional recovery in IWR-1-treated larvae (+ 55% of the swim distance covered by Wnt/β-catenin pathway-inhibited lesioned animals; Fig. 7d). This identifies Col XII as a major downstream effector of Wnt/β-catenin signaling in the regenerating zebrafish spinal cord.

We next asked whether *col12a1a* overexpression is sufficient to augment axon regeneration. To this end, we assessed

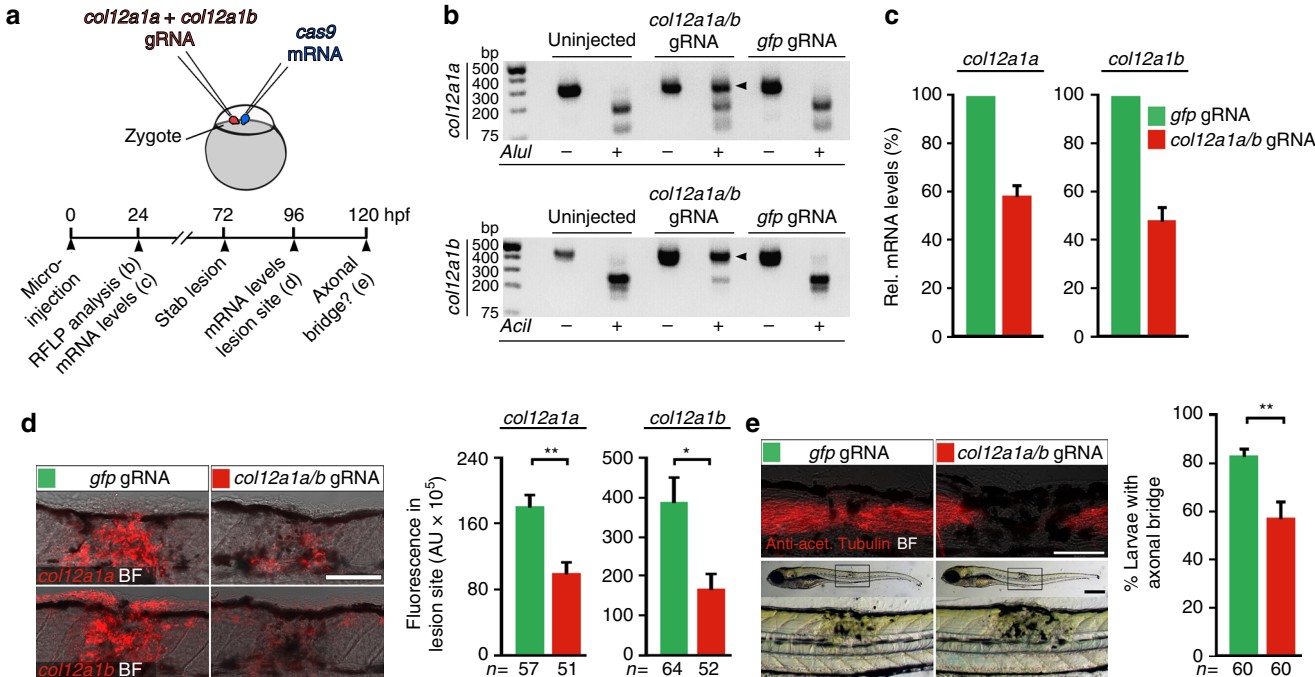

**Fig. 6** CRISPR/Cas9-mediated disruption of *col12a1a/b* leads to depletion of mRNA and impaired axonal regeneration. **a** Cartoon showing strategy used for CRISPR/Cas9-mediated *col12a1a/b* gene disruption. *cas9* mRNA and small guide-RNA (gRNA) targeting *col12a1a* and *col12a1b* were co-injected into the zygote. Crispants were analysed for *col12a1a/b* mRNA levels and axonal regeneration as indicated in the timeline. **b** Restriction fragment length polymorphisms (RFLP) analysis reveals efficient somatic mutation in the gRNA target site (indicated by resistance to restriction endonuclease digest; arrowheads) in both *col12a1* paralogs after microinjection. **c** *col12a1a/b* Crispants exhibit reduced developmental *col12a1a* and *col12a1b* mRNA expression, determined by quantitative RT-PCR. Representative plot from three independent experiments for each mRNA are shown. mRNA levels in *gfp* gRNA-injected control animals were set to 100%. **d** *col12a1a/b* Crispants exhibit reduced *col12a1a* and *col12a1b* mRNA levels after lesion, determined by quantification of fluorescent in situ hybridization signal in the lesion site (*t*-test: **$P < 0.01$, *$P < 0.05$). **e** *col12a1a/b* Crispants show reduced axonal bridging events after lesion (*t*-test: **$P < 0.01$). **a–e** Views are lateral (dorsal is *up*, rostral is *left*). BF: brightfield. *Scale bars*: 200 μm **e** and 100 μm **d**,**e**. *Error bars* indicate s.e.m

axonal bridging in *hs*:col12a1a transgenic animals and their wild type control siblings at 18 hpl (Fig. 7e). At this early time point, 41% of lesioned control larvae showed axonal continuity (anti-acetylated Tubulin⁺) between spinal cord ends. Remarkably, *col12a1a* overexpression increased the proportion of larvae with axonal bridge to 63%. Thus, Col XII is sufficient to enhance axon regeneration after a spinal lesion in zebrafish. Taken together, these findings determine Col XII as major pro-regenerative, Wnt/β-catenin-controlled ECM factor in the spinal lesion site.

## Discussion

Here, we demonstrate an important role for Wnt/β-catenin signaling in controlling the composition of the spinal lesion ECM, which is essential for functional axon regeneration in zebrafish. Our data support a model (Fig. 8) in which spinal cord transection triggers upregulation of Wnt/β-catenin signaling in non-neural cells. The vast majority of these are fibroblast-like cells accumulating in the lesion site. Wnt/β-catenin signaling regulates *col12a1* transcription and deposition of Col XII by fibroblast-like cells in the spinal lesion ECM. The presence of Col XII in the ECM promotes axon growth across the lesion site and functional recovery.

How does Col XII support axonal regeneration? Col XII is abundant in the spinal lesion site in species of high regenerative capacity, such as in larval and adult zebrafish (this report) and salamanders[6]. Col XII is a member of the FACITs family of collagens, which do not form supramolecular aggregates but are associated with Collagen fibrils[46, 52]. Previously, Col XII

was found to interact with Col I fibrils[52]. However, we find that Col I immunoreactivity remained grossly unchanged after interfering with *col12a1a/b* translation. Col XII also interacts with Col VI in forming matrix bridges between developing osteoblasts[53]. Interestingly, we find in our expression screen that *col6a2*, coding for a Col VI α-chain, is co-regulated with *col12a1a/b* by Wnt/β-catenin signaling, which may be related to functional interactions of the proteins. In addition, Col XII binds a variety of other matrix proteins, such as Tenascin-X, Decorin, and Fibromodulin[54, 55]. Hence, Col XII could promote growth of axons indirectly by organising the lesion site ECM. Another possibility is that Col XII could directly interact with axons via specific receptors. This is supported by our observation that overexpression of *col12a1a* alone is sufficient to enhance axonal regeneration. While specific axonal receptors for Col XII have not been identified, potential direct Col XII-axon interactions could be mediated by its integrin-binding site, the NC1 domain, as reported for Col IV[46, 56, 57]. Future studies will need to clarify the precise mechanism by which Col XII promotes axonal re-growth.

While our results identify Col XII as critical factor for axon regeneration, several additional ECM molecules likely contribute to establishing the growth promoting lesion site environment found in zebrafish. For example, Tenascin-C has been shown to be important for spinal cord regeneration in zebrafish, but possible interactions with Col XII are not known[5]. Furthermore, Col I and Fibronectin, which we find to be abundantly present in the lesion site, are also known ECM components that promote axons growth[45, 58, 59].

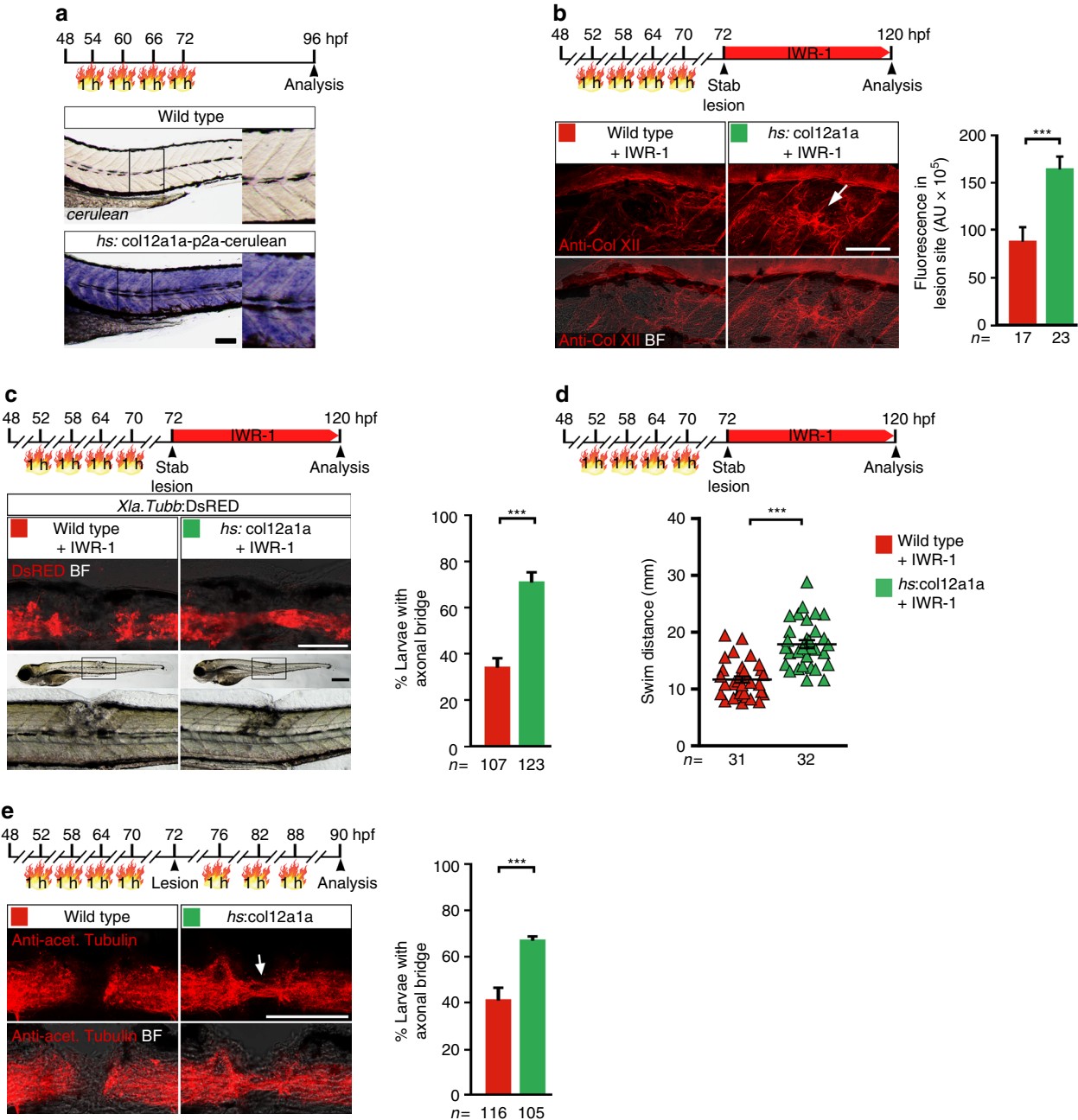

**Fig. 7** Overexpression of *col12a1a* promotes axon regeneration in Wnt/β-catenin pathway-inhibited and wild type zebrafish larvae. **a** *cerulean* mRNA is robustly detectable 24 h after the last heat shock in *hs*:col12a1a-p2a-Cerulean transgenic larvae. **b** *col12a1a* overexpression increases anti-Col XII immunoreactivity in the lesion site of Wnt/β-pathway-inhibited (IWR-1 treatment) *hs*:col12a1a transgenic larvae (*arrow*; *t*-test: ***P < 0.001). **c** *col12a1a* overexpression rescues axon regeneration in Wnt/β-catenin pathway-inhibited (IWR-1 treatment) *hs*:col12a1a transgenic larvae (Fischer's exact test: ***P < 0.001). **d** *col12a1a* overexpression improves functional recovery in Wnt/β-catenin pathway-inhibited (IWR-1 treatment) *hs*:col12a1a transgenic larvae (*t*-test: ***P < 0.001). **e** *col12a1a* overexpression is sufficient to augment axon regeneration (Fischer's exact test: *** P< 0.001). **a–e** Views are lateral (dorsal is *up*, rostral is *left*). BF: brightfield. *Scale bars*: 200 μm **c** and 100 μm **a–c**, **e**. *Error bars* indicate s.e.m

How does Wnt/β-catenin signaling act on *col12a1a/b* expression? Wnt-responding cells co-express *col12a1a/b*. Hence, Wnt/β-catenin pathway activity likely acts in the same cells to increase expression of *col12a1a/b* after a lesion. Regulation of Collagens in fibroblasts by Wnt/β-catenin signaling has also been observed in dermal fibroblasts or myofibroblasts in the context of fibrosis[8–10, 60]. However, a sizable population of fibroblast-like cells showed Wnt-dependent *col12a1a/b* upregulation without having detectable Wnt/β-catenin pathway activity. Our results show that the effect of Wnt/β-catenin

pathway inhibition on *col12a1a/b* expression is delayed and therefore likely indirect and that Wnt/β-catenin signaling does not need to be sustained for *col12a1a/b* expression and successful regeneration. These observations are consistent with the possibility that Wnt/β-catenin signaling is only transiently active in *col12a1a/b*-expressing cells. Alternatively, secondary signals from Wnt-responding cells could instruct transcriptional regulation of *col12a1a/b* in non-Wnt-responsive cells.

Are astroglia-like cell bridges a necessary substrate for axonal regeneration across a spinal lesion site? Regenerating axons could

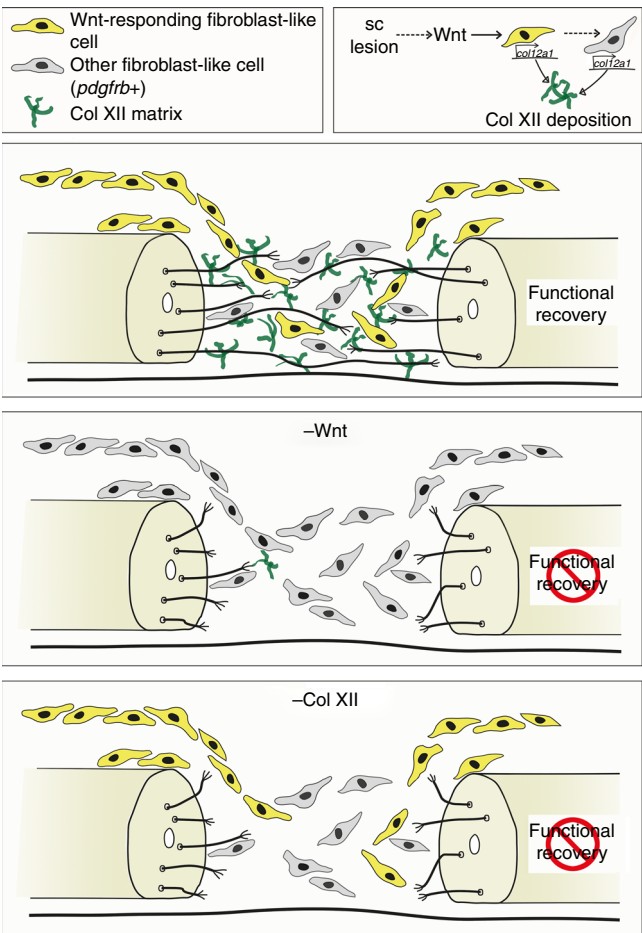

**Fig. 8** Model for Wnt/β-catenin function during axonal regeneration in the zebrafish spinal cord. Spinal cord (sc) lesion triggers Wnt/β-catenin pathway activation in fibroblast-like cells. Wnt/β-catenin signaling controls *col12a1a/b* transcription and deposition of Col XII in the lesion site ECM, through which axons regenerate to facilitate functional recovery. Interference with Wnt/β-catenin signaling in fibroblast-like cells or Col XII deposition prevents axon regeneration and functional recovery

directly interact with the lesion site ECM or simply follow glial processes that bridge the lesion site. In histological preparations this is difficult to assess, as axons and glial processes tend to fasciculate[27]. Indeed, at 2 dpl almost all tissue bridges were composed of neuronal and glial processes. However, simultaneous time-lapse video microscopy imaging of axonal and glial processes indicated that the majority of axons highly dynamically enter and cross the lesion site independently of glial processes. Moreover, our data show that re-establishment of axonal continuity of the transected spinal cord precedes that of glial continuity, similar to observations in the lesioned spinal cord of the adult eel[61]. Finally, in contrast to reducing the amount of Col XII protein in the lesion site, massive depletion of glial processes did not reduce the proportion of animals that showed axonal continuity after regeneration. While these observations support the relative importance of the ECM compared to that of glial substrates for axonal regeneration, we cannot exclude that fine glial processes that escaped detection and ablation still served as a substrate for axon re-growth. Moreover, glial processes are likely to support regeneration by physically aligning with regenerating fascicles[27] and producing essential growth-factors[28], which may be more important in a larger lesion site of adult zebrafish.

Is larval regeneration comparable to adult spinal cord regeneration? In both larvae and adult zebrafish, functional regeneration after complete spinal cord transection depends on axons successfully crossing a non-neural lesion site (this report and refs. [11], [13], [28]) and forming connections[11], [12]. Both show lesion-specific upregulation of *col12a1a* and axons regenerate in close proximity to Col XII in larval and adult zebrafish (this report). At least some axons grow independently of glia at both developmental stages (this report and ref. [27]) and, crucially both larval and adult regeneration depend on Wnt/β-catenin signaling (this report and refs. [14], [15]). This suggests that key aspects of regeneration are conserved across developmental stages. However, regeneration times in adults are much longer than in larvae[62], such that timing of signaling processes may differ in adults.

In conclusion, Col XII deposition, controlled by Wnt-responsive fibroblasts, promotes spinal cord regeneration in zebrafish. Interestingly, transplanting Wnt3a-secreting fibroblasts into a spinal lesion site in rats enhances functional recovery[63]. Thus, elucidating the molecular cues that promote axon regeneration in zebrafish could provide strategies in mammals to modulate the composition of the lesion site to tip the balance towards a growth-conducive ECM environment.

## Methods

**Animals**. All fish were kept and bred in our laboratory fish facility according to standard procedures[64], and all experiments have been approved by the British Home Office (project license no.: 70/8805). Details of the transgenic zebrafish lines used can be found in Supplementary Note 1. We used the WIK wild type strain of zebrafish and the following transgenic zebrafish lines: *6xTCF/Lef-miniP*:2dGFP (*6xTCF*:dGFP[24]); *7xTCF-Xla.Siam*:nlsmCherry[1a5] (*7xTCF*:mCherry[23]); *gfap*:Gal4ff[s995];*UAS-E1b*:Eco.NfsB-mCherry[c264] (*gfap*:Gal4ff[s995];*UAS*:NTR-mCherry[31]);*gfap*:GFP[mi200129], *her4.3*:EGFP[y83] (formerly known as *her4.1*:EGFP[30]); *her4.3*:irtTAM2(3F)-p2a-AmCyan[ulm6] (*her4.3*:TetA AmCyan[25]); *her4.1*:Tet-GBD-p2a-mCherry[tud6] (*her4.3*:mCherry[41]); *hsp70l*:dkk1-GFP[w32] (*hs*:dkk1[42]); *hsp70l*:Mmu.Axin1-YFP[w35] (*hs*:Axin1[34]); *hspl70l*:wnt8a-GFP[w34](*hs*:wnt8[47]); *krtt1c19e*:EGFP[sq1744], *pdgfrb*:Gal4ff[ncv24];*UAS*:GFP (*pdgfrb*:GFP[48]) *TetRE*:Mmu.Axin1-YFP[tud1] (*TetRE*:Axin1-YFP[41]); *Top*:dGFP[w2522]; *ubiquitin*:irtTAM2(3F)-p2a-AmCyan[ulm2](*ubi*:TetA AmCyan[25]);*Xla.Tubb*:DsRED[zf148;26]. The *lef1*:irtTAM2(3F)-p2a-AmCyan[ulm11] (*lef1*:TetA AmCyan), *Xla.Tubb*:irtTAM2(3F)-p2a-AmCyan[ue103] (*Xla.Tubb*:TetA AmCyan) and *hsp70l*:col12a1a-p2a-Cerulen[ue104] (*hsp70l*:col12a1a) transgenic zebrafish lines were established by Tol2-mediated transgenesis[65] using the DNA constructs described below.

**DNA constructs**. The *lef1*:irtTAM2(3F)-p2A-AmCyan construct was generated by replacing the translation start codon of *lef1* in the BAC clone CH73-164H6 with the previously described irtTAM2(3F)-p2A-AmCyan TetActivator cassette[41], [66] using standard BAC recombination techniques[67]. Additionally, Tol2 inverted repeats were introduced into the BAC backbone. The final BAC was purified using the HiPure Midiprep kit (Invitrogen). The *Xla.Tubb*:irtTAM2(3F)-p2a-AmCyan construct was generated by cloning the previously described sequence of the *Xenopus laevis* neural-specific beta tubulin regulatory element (*Xla.Tubb2b*) upstream of the published doxycycline (DOX)-inducible transcriptional activator [irtTAM2(3F)] tagged with p2a and AmCyan[26], [41], [66]. For generation of the *hsp70l*:col12a1a-p2a-Cerulean construct, 3 overlapping fragments spanning the full-length *col12a1a* CDS were amplified from 1 dpl cDNA (primer sequences are given in Supplementary Table 1) and fused via internal restriction sites (BsiWI, BsrGI). The complete *col12a1a* CDS was then subcloned into the *hsp70l*:irtTAM-p2a-Cerulean construct[41] thereby replacing the irtTAM TetActivator CDS.

**Reverse transcriptase PCR and quantitative RT-PCR**. To assess the effect of IWR-1 treatment on lesion-induced gene expression total RNA from lesion site-enriched trunk tissue was extracted using Trizol (Invitrogen). Twenty larvae were pooled for each experimental condition. To assess the effect of *col12a1a/b* gRNA/cas9 mRNA microinjection on developmental *col12a1a/b* mRNA levels total RNA from ten 24 hpf embryos was extracted using Trizol. Reverse transcription of 500 ng RNA was performed with the SuperScript III kit (Invitrogen) using a combination of oligo(dT) and random hexamer primer. Standard RT-PCR (58 °C, 30 cycles) was performed using 10 mM of each dNTP and 10 μM of each primer. qRT-PCR was performed at 58 °C using Roche Light Cycler 96 and relative mRNA levels determined using Roche Light Cycler 96 SW1.1 software. Samples were run in triplicate and expression levels were normalized to β-actin control. Primer were designed to span an exon–exon

junction using Primer-BLAST software. Primer sequences are given in Supplementary Table 2.

**Spinal cord lesions and behavioral recovery**. Zebrafish larvae (3 dpf) were anesthetized in PBS containing 0.02% aminobenzoic acid ethyl methyl ester (MS222; Sigma-Aldrich). For dorsal incision lesions, larvae were transferred to a lid of a plastic Petri dish. Following aspiration of excess water, which placed the larvae in a lateral position, the tip of a sharp 30 G syringe needle was used to inflict a lesion in the dorsal trunk at the level of the anal pore. Survival of larvae was typically > 95%. Larvae that had undergone extensive damage to the notochord showed the formation of a notochord-derived bulgy tissue structure (~ 10%). Those larvae were excluded from further analysis. For less invasive stab lesions, larvae were transferred to an agarose-coated Petri dish and larvae were placed in a lateral position through removal of excess water. The tip of a sharp 30 G syringe needle was pushed through the spinal cord at the level of the anal pore immediately dorsal to the notochord. Kinetics of anatomical/functional spinal cord regeneration were comparable in larvae that underwent dorsal incision lesion or stab lesion. After surgery, larvae were returned to E3 medium for recovery and kept at 28.5 °C. Wherever possible, *Xla.Tubb*:DsRED transgenic larvae were used for experiments to visually verify completeness of spinal cord transection after lesion. Analysis of behavioral recovery was performed as described previously[13]. Behavioral data are shown as distance travelled within 15 s after touch, averaged for triplicate measures per larvae. Adult spinal cord lesions have been described previously[62]. Briefly, fish ($\geq$ 3 month post-fertilization) were anesthetized by immersion in 0.02% MS222 in PBS for 5 min. A longitudinal incision was made at the side of the fish to expose the vertebral column. The spinal cord was completely transected under visual control, 3.5 mm caudal to the brainstem-spinal cord junction.

**Drug treatments and heat-shock treatments**. Heat shocks and drug treatments were performed according to the schematic timelines shown with each experiment. For heat shocks, larvae were kept in 50 ml conical centrifuge tubes filled with 40 ml E3 embryo medium, which floated in a programmable thermostat-controlled water bath (Lauda, Germany). Heat shocks of transgenic animals and non-transgenic sibling controls were performed for 1 h at 38 °C after which larvae were returned to 28.5 °C. For drug treatments, up to 10 larvae were incubated in 7 ml of E3 embryo medium, containing the drug. Larvae were kept in the dark at 28.5 °C and water was exchanged daily. The Axin1 stabilizer IWR-1 (Sigma-Aldrich) was dissolved in DMSO (6 mM stock) and used at a final concentration of 15 μM. Doxycycline (DOX; Sigma-Aldrich) was dissolved in 50% EtOH (50 mg/ml stock) and used at a final concentration of 40 μg/ml. For targeted cell ablation, the pro-drug Metranidozole (MTZ; Sigma-Adrich) was dissolved in DMSO (200 mM) and used at a final concentration of 2 mM.

**Morpholino injection**. Knockdown of *col12a1a/b* was performed by injecting Vivo-Morpholinos (MO) to *col12a1a/b*, which act as cell-permeable translation blocking anti-sense molecules, into the circulation for efficient distribution in 3 dpf larvae (Supplementary Fig. 11a[50]). Two translation blocking Vivo-Morpholinos, antisense to the transcriptional start site of *col12a1a* (ENSDART00000154728) and *col12a1b* (ENSDART00000025926), were designed (Gene Tools, Philomath, OR, USA): *col12a1a* MO 5′-CGGCCAAAGA-CAACCTGATCTTCAT-3′; *col12a1b* MO 5′-TGCCAAATGCCTGACCGA-CATCTTC-3′. 100% homology of MO sequences to target sites were confirmed by sequencing. Approximately a total of 27 nl injection mix (9 repetitive injections 3 nl each), containing Cascade Blue Dextran (MW: 3000; Molecular Probes) and 0.25 mM of each Morpholino, was injected directly into the ventricle of 3 dpf larvae as described previously[50]. Larvae that showed strong and ubiquitous fluorescence at 2–3 h post-injection were subsequently lesioned. As control a standard Vivo-Morpholino from Gene Tools was used.

**CRISPR-mediated genome editing**. CRISPR *col12a1a* and *col12a1b* gRNA were designed using a combination of different webtools: ZiFit (http://zifit.partners.org/ ZiFiT), CRISPR Design (http://crispr.mit.edu), and Mojo Hand (http://talendesign. org). gRNA expression vectors were built by ligation of annealed oligonucleotides into pT7-gRNA expression vector (Addgene #46759) as described previously[51]. Capped sense *nls-zCas9-nls* RNA was synthesized using mMESSAGE mMACHINE T3 kit (Ambion) and pT3TS-nCas9n (Addgene # 46757) as template[51]. RNA was purified using RNAeasy Mini Kit (Qiagen). gRNA was in vitro transcribed from gRNA expression vectors using mMESSAGE mMACHINE T7 kit and purified using mirVana miRNA isolation kit (Ambion). All synthesized RNAs were assessed for size and quality by gel electrophoresis. A mix of 75 pg gRNA and 150 pg *nls-zCas9-nls* RNA was microinjected into one-cell-stage embryos to determine efficiency of individual gRNAs to introduce mutations in the target site in pools of 24 hpf embryos using restriction fragment length polymorphisms (RFLP) analysis. For subsequent experiments we used a combination (75 pg each) of *col12a1a* gRNA#1 (target sequence 5′-GGCTGTGGTTCAGTACAGCT-3′; targeting exon 6) and *col12a1b* gRNA#1 (target sequence 5′- GGTCAGGCATTTGGCAGCGG-3′; targeting exon 2) and 150 pg *nls-zCas9-nls* RNA for microinjection. A previously described gRNA targeting GFP (target sequence: 5′GGCGAGGGCGATGC-CACCTA-3)[68] served as control. Efficient mutagenesis of target loci was confirmed

after each injection by RFLP analysis as follows. For *col12a1a* gRNA#1, a 308 bp fragment was amplified using primers 5′- TGGAGTGTGGGAAGAGAAAACTT-3′ and 5′- CTAATGAGAATTTGTCGGCAGCG-3′ and digested with AluI. For *col12a1b* gRNA#1, a 400 bp fragment was amplified using primers 5′-TGGAG-CATGTATTTTCCCCTTGA-3′ and 5′-GCTCCAGTCCTTTTGTTCATTCC-3′ and digested with AciI.

**Live imaging of zebrafish larvae and time-lapse imaging**. For live confocal imaging, zebrafish larvae were anesthetized in PBS with 0.02% MS222 and mounted in the appropriate orientation in 1% low melting point agarose (Ultra-Pure^TM, Invitrogen). During imaging the larvae were covered with 0.01% MS222-containing fish water to keep preparations from drying out. For time-lapse imaging, agarose covering the lesion site was gently removed after gelation. Time-lapse imaging was performed for 19 h starting at 6 hpl. After initial visual inspection, acquired time-lapse images were denoised to improve image quality (enhance detail). Background noise, caused by the low signal to noise ratio, was removed from acquired time-lapse images using the ImageJ plugin CANDLE-J algorithm, which allows to remove the noise compartment while preserving the structural information with high fidelity[69]. We verified that this procedure did not remove signals from processes of low fluorescence intensity by comparing raw movies with CANDLE-J-processed movies, which showed that that edges of features remained conserved after denoising.

**Quantifications and statistics**. Re-established axonal or glial connections (bridges) were scored in static preparations (fixed immunolabelled samples, live transgenic animals) or in still images of time-lapse movies at time points of interest. Larvae were directly visually evaluated using a compound fluorescence microscope (Zeiss Imager.Z1), or after confocal imaging (Zeiss LSM 710, 880). A larva was scored as having a bridged lesion site when continuity of the neuronal or glial labeling between the rostral and caudal part of the spinal cord was observed. Continuity of labeling was defined as at least one fascicle being continuous between rostral and caudal spinal cord ends irrespective of the fascicle thickness. Larvae in which the lesion site was obscured by melanocytes were excluded from analysis. For all quantifications, the observer was blinded to the experimental treatments. Scores of key experiments were validated by a second independent observer.

Quantification of axonal and glial fascicle composition in static preparations at 1 dpl was performed after confocal imaging (Zeiss LSM 880) of immunolabeled or live animals. Fascicles were identified as fluorescent protrusions from the severed spinal cord that projected into the injury site for at least 20 μm or had crossed it completely. This was done by inspecting single optical sections and 3D renderings of the lesion site. If more than half of the length of a fascicle contained fluorescence for both glial and neuronal markers it was scored as "mixed", otherwise it was scored as purely "axonal" or "glial" depending on the markers used. The threshold of half the length was used to exclude situations in which one type of process followed the other long after that one was established.

For quantification of axonal and glial fascicle composition in time-lapse movies we included only fascicles that grew during the observation period.

Fascicles were classified as purely "axonal" or "glial" when the distal part of a given fascicle was labelled by the respective marker throughout the entire observation period. In "mixed" fascicles, both markers were present in the distal portion of the fascicle.

Fascicles were considered to be in close apposition with ECM in histological preparations when > 50% of an axonal fascicle that had entered or crossed the lesion site was aligned with ECM immunoreactivity. This was assessed in single optical sections and 3D renderings.

Quantification of immunohistochemistry or fluorescent in situ hybridization signal in the lesion site was performed on captured images of whole mount samples using ImageJ software as described in subsection "Image analysis". Samples were imaged using a compound fluorescence microscope (Zeiss Imager.Z1) or confocal microscope (Zeiss LSM 710, 880). Measurements were performed in a blinded fashion.

Quantification of GFP⁺/p63⁺ and GFP⁺/p63⁻ cells in 1 dpl and 2 dpl *6xTCF*: dGFP transgenic larvae was performed on immunolabeled sections of individual animals ($n = 10$).

For non-quantitative data, the number of specimens that showed a given phenotype and the total number of specimens analyzed is given as a ratio in each figure. Phenotypes were scored in a blinded fashion.

All experiments were performed at least three times. Animals were randomly assigned to different experimental groups but no formal method of randomisation was used. Power analysis of pilot experiments informed minimum samples size. All quantitative data were tested for normality and analyzed with parametric and non-parametric tests as appropriate. An F-test was used to check for equal variances. We used one-way ANOVA followed by Dunn's multiple comparison test, two-way ANOVA, followed by Student's *t*-test, or Fischer's exact test, as indicated in the figure legends. *$P < 0.05$, **$P < 0.01$, ***$P < 0.001$, n.s. indicates not significant. Error bars always indicate the standard error of the mean (SEM).

**Image analysis**. Quantitative analysis of immunohistochemistry or fluorescent ISH signals in the lesion site was performed on captured images. Samples were

imaged using a compound fluorescence microscope (Zeiss Imager.Z1) or confocal microscope (Zeiss LSM 710, 880). Analysis of confocal images were performed on maximum intensity projections that were generated from confocal stacks. Image analysis was based on published protocols for fluorescent signal quantification[70]. For all measurements except in Fig. 7b, ImageJ software was used to determine the number of pixel above an intensity threshold (thresholded pixel area) in the region of interest (ROI) (an example of such a measurement is shown in Supplementary Fig. 14). In brief, a minimum intensity threshold was applied to all raw images from the same dataset, limiting pixels to only those of equal or higher intensity. The intensity threshold was determined for each experimental data set individually, using untreated lesioned control samples as reference. The threshold was set such that the thresholded pixel area included most pixels of the fluorescence signal domain in the lesion site while low background signal in the vicinity of the lesion site was largely excluded (efficient image segmentation). After thresholding, a ROI was additionally defined manually to exclude pixels from analysis that fell within the threshold intensity range but were located outside the lesion site. This was necessary in some samples to exclude autofluorescence (e.g., from blood vessels) or constitutively labelled domains (e.g., myosepta for Col XII). Measurements of immunoreactivity in the lesion site, shown in Fig. 7b, was performed by determining the pixel area in a pre-defined ROI of constant size without prior thresholding. This was necessary due to low signal to noise ratio. In brief, the image was converted to a binary image using the automated binary thresholding function of ImageJ. The ROI was placed in the center of the lesion site. All measurements were performed in a blinded fashion.

**In situ hybridization**. All incubations were performed at room temperature unless stated otherwise.

For chromogenic and Fast Red fluorescence whole mount in situ hybridization (ISH), larvae were fixed overnight at 4 °C in 4% PFA-PBS. On the following day, larvae were washed twice in PBT (0.1% Tween-20 in PBS) and incubated for 30 min in PBT containing 40 µg/ml Proteinase K (Invitrogen). Thereafter, larvae were washed briefly in PBT and were refixed for 20 min in 4% PFA-PBS followed by five washes in PBT for 5 min each. Thereafter, larvae were incubated at 65 °C for > 1 h in pre-warmed hybridization buffer (5x SSC, 500 µg/ml type VI Torula yeast RNA, 50 µg/ml Heparin, 0.1% Tween 20, 9 mM citric acid [Monohydrate]), 50% deionized formamide, pH 6.0). Hybridization buffer was replaced with digoxigenin (DIG) and/or fluorescein-labelled ISH probes diluted in hybridization buffer and incubated at 65 °C overnight. On the following day, larvae were washed at 65 °C once in hybridization buffer, three times in 50% 2×SSCT/50% deionized formamide, and twice in 2xSSCT for 20 min each, followed by four washes at 65 °C in 0.2x SSCT (30x SSC: 300 mM NaCl, 200 mM Na-Citrate, pH 7; SSCT: 0.1 % Tween 20 in 1xSSC) for 30 min each. Thereafter, larvae were washed twice in PBT and incubated for > 1 h in blocking buffer (5% heat-inactivated sheep serum, 10 mg/ml BSA in PBT) for 5 min each under slow agitation. Subsequently, larvae were incubated overnight at 4 °C in blocking buffer containing pre-absorbed anti-DIG or anti-fluorescein antibody coupled to alkaline phosphatase (1:3000–4000). On the following day, larvae were washed six times in PBT for 20 min each, followed by three washes in staining buffer (50 mM MgCl₂, 100 mM NaCl, 100 mM Tris-HCl, 0.1% Tween 20, pH 9.5) for 5 min each. Color reaction was performed by incubating larvae in staining buffer supplemented with NBT/BCIP (Sigma-Aldrich) or SIGMAFAST Fast Red (Sigma-Aldrich) substrate. The staining reaction was terminated by washing larvae in PBT. For chromogenic ISH, background staining was cleared by incubating larvae two times in 100% EtOH for 15 min each and once in 50% EtOH-PBT for 5 min, followed by five washes in PBT for 5 min each.

TSA fluorescence ISH was performed using the TSA Plus System Kit (Perkin Elmer) as described for chromogenic ISH with some modifications. 1) Proteinase K treatment was performed for 20 min (20 µg/ml Proteinase K). 2) Before pre-hybridization, endogenous peroxidase activity was quenched by incubating larvae in 1% H₂O₂ for 20 min. 3) Larvae were blocked in blocking reagent (1% in PBT) provided with the kit. 4) Antibodies used to detect labelled RNA hybrids were anti-digoxigenin or anti-fluorescein antibodies coupled to peroxidase (1:500). 5) Color reaction was performed for 30 min in amplification buffer containing Tyramid substrate (1:500) provided with the Kit, followed by washes in PBT for 20 min each.

Simultaneous detection of two different mRNAs by ISH was performed using two-color TSA fluorescence or combinations of Fast Red and TSA fluorescence. Stained samples (Fast Red and TSA fluorescence) were embedded in 4% Agarose-PBS and sectioned at 50–100 µm using a vibratome. Sections were stained for DAPI (Thermo Scientific) to visualize nuclei, followed by two washes in PBS and mounted in 75% Glycerol-PBS. Larvae were incubated for 30 min in 2% H₂O₂ after the first color reaction to deactivate antibody-conjugated peroxidase.

For Fast Red fluorescent ISH on floating agarose sections of larval zebrafish, larvae were fixed in 4% PFA-PBS at 4 °C overnight. After two brief washes in PBT (0.1% Tween-20 in PBS), larvae were embedded in 4% Agarose-PBS and sectioned at 50–100 µm using a vibratome. Sections were incubated for 20 min in PBT containing 10 µg/ml Proteinase K (Invitrogen) and further processed as described for whole mount preparations.

For Fast Red fluorescent ISH on floating agarose sections of adult zebrafish tissue, zebrafish were transcardially perfused with PBS followed by fixative (4% PFA). The lesion site with surrounding non-neural tissue and spinal cord was dissected in a semi-intact preparation and the tissue was post-fixed in 4% PFA

overnight. Tissue was embedded in 4% Agarose-PBS and sectioned at 100–200 µm using a vibratome. Sections were briefly rinsed in PBS, incubated for 30 min in PBT containing 40 µg/ml Proteinase K (Invitrogen) and further processed as described for whole mount preparations.

To detect *gfp* transcripts in *6xTCF*:dGFP;*hs*:Axin1-YFP double transgenic larvae we used a probe against the destabilizing signal (sequence coding for residues 422–461 of mouse ornithine decarboxylase [MODC] as reported previously[25]).

To detect *cerulean* transcripts in *hs*:col12a1a-p2a-Cerulean transgenic fish we used a probe against the closely sequence-related *gfp* as described previously[25].

Information on ISH probes, including primer sequences used for molecular cloning, is provided in the Supplementary Data 1. Where possible, probes to detect collagen chains-coding transcripts were designed to include 3′UTR, which in most cases contained sequences with the lowest possible similarity to other collagen genes. During the ISH screen only genes were considered detectably expressed for which probes showed a signal in the lesion site within 8 h after initiation of staining reaction. Functionality of probes that showed no staining in the lesion site was verified by the presence of specific staining domains in non-lesion site tissue, which was true for all probes except *col4a4*, *col8a1b*, and *col16a1*, which did not show constitutive expression domains in 4 dpf larvae. Lesion-induced upregulation of genes in the lesion site was determined by comparing the signal intensity in the lesion site to that in adjacent unlesioned trunk tissue.

**Immunofluorescence**. All incubations were performed at room temperature unless stated otherwise.

For whole mount acetylated Tubulin/GFAP immunolabeling, larvae were fixed in 2% TCA-PBTx (1% Triton X-100 in PBS) for 3 h, followed by brief washes in PBTx and PBS. Thereafter, larvae were incubated for 30 min in 0.25% Trypsin (Gibco) diluted in PBS, followed by five washes in PBTx for 5 min each. Larvae were blocked in 4% BSA-PBTx for 1 h and incubated with primary antibody (1:300) diluted in blocking buffer at 4 °C overnight. On the following day, larvae were washed six times in PBTx for 20 min each, followed by incubation with secondary antibody of interest diluted in blocking buffer (1:300) at 4 °C overnight. On the following day, larvae were washed six times each in PBTx for 20 min each, followed by mounting in PBS or 75% Glycerol-PBS.

For whole mount GFP/acetylated Tubulin/p63 immunolabelling in *6xTCF*: dGFP transgenic larvae, larvae were fixed in 4% PFA-PBS containing 1% DMSO at 4 °C overnight. On the following day, larvae were washed four times in PBS, followed by two washes in PBTx (0.2% Triton X-100 in PBS) for 5 min each. Thereafter, larvae were permeabilized by incubation in PBS containing 2 mg/ml Collagenase (Sigma-Aldrich) for 25 min. To terminate Collagenase digest, larvae were briefly rinsed in PBTx. Subsequently, larvae were incubated in 50 mM glycine in PBTx for 10 min, followed by a brief rinse in PBTx. Larvae were blocked in blocking buffer (0.7% PBTx, 1% DMSO, 1% normal donkey serum, 1% BSA) for 2 h, and incubated with primary antibody (1:300–1:500) diluted in blocking buffer at 4 °C overnight. On the following day, larvae were washed three times in PBTx for 15 min each, followed by incubation with secondary antibody of interest diluted in blocking buffer (1:300) at 4 °C overnight. On the following day, larvae were washed three times in PBTx and once in PBS for 15 min each, before mounting in 75% Glycerol-PBS.

For whole mount acetylated Tubulin/GFAP immunolabelling, as well as all other combinations of immunolabelling on whole mounts (e.g., acetylated Tubulin/ Col XII), larvae were fixed in 4% PFA-PBS for 1 h, followed by brief washes in 0.1% PBT (0.1% Tween-20 in PBS). Thereafter, larvae were stepwise dehydrated by successive incubation in 25% MeOH-PBT, 50% MeOH-PBT, 75% MeOH-PBT, 100% MeOH for 5 min each, and stored at −20 °C until need. When required, head and tail of larvae were removed and lesion site-enriched trunk tissue was incubated in 100% MeOH at −20 °C overnight. On the following day, samples were rehydrated by successive incubations in 75% MeOH-PBT, 50% MeOH-PBT and 25% MeOH-PBT for 5 min each, followed by four washes in PBT for 5 min each. Larvae were washed twice in distilled H₂O for 5 min each, followed by incubation in 100% Acetone (pre-chilled to −20 °C) at −20 °C for 10 min. Thereafter, larvae were washed twice in distilled H₂O for 5 min each, followed by two 5 min washes in PBT. Subsequently, larvae were incubated in PBT containing 10 µg/ml Proteinase K (Invitrogen) for 15 min, followed by two brief washes in PBT and re-fixation in 4% PFA-PBS for 15 min. Thereafter, larvae were washed tree times in PBTx (1% Triton X-100 in PBS) 10 min each and incubated in blocking buffer (4% BSA in PBTx) for 1 h. Thereafter, larvae were incubated with primary antibody (1:300) diluted in blocking buffer at 4 °C for 2–3 days. Thereafter, larvae were washed eight times in PBTx for 20 min each, followed by incubation with secondary antibody of interest diluted in blocking buffer (1:300) at 4 °C for 1–2 days. Thereafter, larvae were washed six times in PBTx for 20 min each and once in PBS for 15 min before mounting in 75% Glycerol-PBS.

For immunohistochemistry on floating agarose sections of larval zebrafish, larvae were fixed in 4% PFA-PBS containing 1% DMSO at 4 °C overnight. After two brief washes in PBT (0.1% Tween-20 in PBS), larvae were embedded in 4% Agarose-PBS and sectioned at 50–100 µm using a vibratome. Sections were briefly rinsed in PBS and incubated in PBS containing 2 mg/ml Collagenase (Sigma-Aldrich) for 20 min. Thereafter, sections were washed several times in PBTx (0.5% Triton X-100 in PBS) followed by incubation in blocking buffer (0.7% PBTx, 1% DMSO, 1% normal donkey serum, 1% BSA) for 1 h, and incubated with primary antibody (1:50–1:200) diluted in blocking buffer at 4 °C for 2–3 days.

Thereafter, sections were washed eight times in PBTx for 20 min each, followed by incubation with secondary antibody of interest diluted in blocking buffer (1:300) at 4 °C overnight. On the following day, larvae were washed six times in PBTx and twice in PBS for 20 min each. Sections were stained for DAPI (Thermo Scientific) to visualize nuclei, followed by two washes in PBS and mounted in 75% Glycerol-PBS.

For immunohistochemistry on floating agarose sections of adult zebrafish tissue, zebrafish were transcardially perfused with PBS followed by fixative (4% PFA-PBS). The lesion site with surrounding non-neural tissue and spinal cord was dissected in a semi-intact preparation and the tissue was post-fixed in 4% PFA for 1 h. Tissue was embedded in 4% Agarose-PBS and sectioned at 100–200 μm using a vibratome. Sections were briefly rinsed in PBS and washed several times in PBTx (1% Triton X-100 in PBS) for a total of 1 h followed by incubation in blocking buffer (4% BSA in PBTx). Thereafter, sections were incubated with primary antibody (1:250) diluted in blocking buffer at 4 °C for 3 days. Thereafter, sections were washed eighteen times in PBTx for 10 min each, followed by incubation with secondary antibody of interest diluted in blocking buffer (1:250) at 4 °C for 2 days. On the following day, sections were washed twelve times in PBTx and twice in PBS for 20 min each. Sections were mounted in 75% Glycerol-PBS.

All primary antibodies used are given in Supplementary Table 3. Primary antibodies used are specific for zebrafish antigens as shown in the indicated publications. To detect basal keratinocytes, we used anti-p63 immunohistochemistry, which is a commonly used marker for this cell lineage[37]. The antibody used in this study, anti-tp63 antibody (Sigma-Aldrich), replicates selective labelling of monolayered basal keratinocytes found in previous publications with other antibodies to this antigen. Secondary fluorophore-conjugated antibodies were from Jackson Immuno Research.

**Data availability**. The authors declare that all data supporting the findings of this study are available within the article and its supplementary information files, or from the corresponding authors on reasonable request.

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

## Acknowledgements

We thank Silvere Santos for excellent fish care, Dr. Florence Ruggiero for antibodies, Dr. Bertrand Vernay for technical support, and Drs Didier Stainier, Francesco Argenton, Richard I. Dorsky, Thomas J. Carney, Ajay B. Chitnis, Pamela A. Raymond, Francesca Peri, Tohru Ishitani and Naoki Mochizuki for transgenic fish lines. This research was supported by the Deutsche Forschungsgemeinschaft (Forschungsstipendium WE5736/1-1) to D.W., and the BBSRC (BB/L021498/1) and NC3Rs (NC/l001063/1) to C.G.B. and T.B.

## Author contributions

Conceptualization, D.W., C.G.B., and T.B.; Investigation, D.W., T.M.T., and A.M.; Writing: D.W., C.G.B., and T.B.; Resources: C.H., M.M.R., and G.W.

## Additional information

**Competing interests:** The authors declare no competing financial interests.

