## [Peer Review File · Nature Communications]

Reviewers' Comments:

Reviewer #1 (Remarks to the Author)

This manuscript by Wehner, D. et al. showed that Wnt signal-dependent Collagen XII expression is necessary for axonal regeneration. Using zebrafish larvae the authors found that spinal cord transection induces Wnt signal activity in fibroblast-like cells and the activity is pivotal for axonal growth and functional recovery. Wnt signal increased the expression of Collagen XII in the lesion site, and its downregulation suppressed axonal growth and functional recovery. Taken together with the observation that glial bridges were not an obligatory substrate for dynamic axonal growth, the authors concluded that the lesion site environment regulated by Wnt signaling is important for regeneration of injured spine.

Experimental designs are elegant, and the obtained results are solid. As the authors described, Wnt signaling is known to control fibroblast biology and ECM deposition, and the lesion site ECM promotes spinal cord regeneration in fish. Therefore, the novelty in this study was to identify Collagen XII as a Wnt signaling downstream molecule and to show that Collagen XII is necessary for axonal regeneration by loss of function experiments.

The following experiments are required to strengthen the authors' conclusion.

1. In Fig 1A-C, the authors showed that initially formed axonal and glial fascicles are independently distributed. In addition, axonal connections can be formed independently of glial connection. From these observations, they claim that axonal growth cones extend into lesion sites independently of glial process formation. However, it is not clear whether axonal growth occurs independently of astrocytes per se. The authors can address this issue by depleting astrocytes.
2. The authors need to provide more details on exactly how images were scored in Fig. 1C.
3. The authors claim "The majority of fascicles that entered the lesion site were composed of only axons (49-53%), some fascicles contained both glial and axonal processes (20-21%) and some pure glial fascicles were observed (27-30%)". However, it is not clear how they count neural and glial fascicles.
4. The authors showed that *col1a2* and *fn1a* are expressed in the lesion site in Fig. 1D and that the expression levels of these mRNAs are not reduced by IWR-1 treatment. To address the involvement of Collagen I and Fibronectin 1 in axonal regeneration, the effect of reduction of these proteins by MO should be tested.
5. The important finding of this study is that Wnt signal-induced Collagen XII is required for spinal cord regeneration. Is *col12a1a/b* a direct target gene of Wnt signaling? If so, the authors should examine whether TCF/LEF binds to the *col12a1a/b* promoter region by ChIP assay and perform its promoter analysis. Otherwise, the underlying mechanism that Collagen XII expression is induced by Wnt signaling must be clarified.
6. Co-localization of axonal processes and ECM shown in Fig. 1E and Fig. 4G is partial. It is hard to understand axons navigate along ECM. More precise description and quantification are required.
7. In Fig. 4G the time course of appearance of disappearance of *col12a1a/b* after injury should be shown. Is the association between newly formed axon and Collagen XII observed at early time point such as at 0.5 dpl? Are 6 x TCF:dGFP-expressing cells present in the lesion site at 0.5 dpl? One would like to know whether the close contact of spinal axon and Collagen XII is important for

regeneration.

8. The authors suggested that Wnt signal may not have to be sustained for transcription of col12a1a/b in Fig. 4E. The authors need to confirm that IWR-1 does not affect col12a1a/b expression after 1 dpl.

9. Figs 5 and 6 demonstrated that col12a1a/b expression is required for axonal regeneration using MO and CRISPR/cas9. Gain of function experiments are required to determine the functional importance of Collagen XII. Does overexpression of Collagen XII promote axonal regeneration? Is it possible to rescue the phenotypes of col12a1a/b-depleted larvae by expression of col12a1a/b?

Reviewer #2 (Remarks to the Author)

In this manuscript by Wehner et al the authors investigate the role of Wnt/b-cat signaling and Col12a in CNS regeneration. The authors demonstrate that following spinal cord transection in zebrafish larvae Wnt/b-cat signaling is up-regulated in fibroblast like cells at the lesion site, and that blocking Wnt/b-cat signaling greatly reduces spinal cord regeneration. The authors then identify Col12a as a downstream component of Wnt/b-cat signaling, and provide evidence that like Wnt/b-cat signaling Col12a is also required in vivo for spinal cord regeneration. Overall, this is a very nice study that takes full advantage of the zebrafish as a model to identify genes that promote spinal cord regeneration, providing molecular insights into this process currently not feasible in mammals. Most claims (see below) are supported by solid experimental data, often using several approaches/ reagents to confirm findings. Finally, this study will be of great interest to the field of CNS regeneration, including mammalian CNS regeneration as it identifies a transcriptional target for Wnt/b-cat signaling, Col12a, that promotes spinal cord regeneration.

Main Points

1. From their live cell imaging the authors conclude "that glial bridges are not an obligatory substrate for axon growth across the non-neural lesion site." Given the longstanding debate in the field whether axons or glia first cross the injury site, the authors need to substantiate their initial findings. Glial processes just like axonal growth cones can be of very small diameter and hence hard to detect when cytoplasmic GFP is used. Thus, to substantiate their findings that glial bridges are not an obligatory substrate it appears imperative to conduct this experiment using membrane tagged markers for glia processes, or electron microscopy. Otherwise this statement should be softened to reflect the possibility that while not detected here glial process might well be present when regenerating axons traverse the injury site.

2. The authors examine expression of 44 zebrafish collagens after spinal cord injury and report that only Col6 and Col12a (1a, 1b) are down regulated when Wnt/b-cat signaling is inhibited. It was less clear which of the 42 collagens were up-regulated after spinal cord injury (without IWR-1 treatment), and specifically whether Col12(1a, 1b) are indeed up-regulated following injury.

3. The authors provide compelling evidence that Col12a/b is up-regulated in a subset of Wnt/b-cat responsive cells and that forced Wnt8 expression (via *hs:wnt8*) enhances Col12a (1a, 1b) expression levels at the lesion site. Combined with the observation that reducing Col12a function recapitulates the phenotype (i.e. reduced regeneration) observed when blocking Wnt/b-cat signaling, this suggests that Wnt/b-cat signaling is required to activate Col12a (1a, 1b) expression. It therefore seems important to determine whether Col12a expression is sufficient to restore spinal cord regeneration in larvae in which Wnt/b-cat signaling is blocked. This 'rescue' experiment would complement the evidence for Wnt/b-cat inducing Col12 (a, b) expression gained mostly by overexpression (e.g. Axin, *wnt8*) analysis, and would provide some mechanistic insights into whether Col12A (1a, 1b) are the major Wnt/b-cat targets relevant for spinal cord regeneration.

4. The authors show that Wnt/b-cat signaling is important in vivo for spinal cord regeneration.

However, which step during the complex process of regeneration Wnt/b-cat signaling promotes remains unclear. For example, does Wnt/b-cat signaling initiate and/or sustain regeneration? By inducing Axin expression or starting IWR-1 treatment later and later time points post injury, it should be possible to determine the first time point when Wnt/b-cat signaling is required (i.e. the first time point at which blocking Wnt/b-cat signaling no longer affects regeneration). This additional data would provide important mechanistic insights into the role of Wnt/b-cat signaling in spinal cord regeneration.

5. Finally, without stable Col12a (1a, 1b) mutant lines it seems important to correlate the CRISP analysis on a larvae by larvae basis with the regeneration assay. Specifically, on a larvae by larvae basis what is the extent of regeneration with none, only col12a, only col12b or col12a/b bi-allelic inactivation?

Reviewer #3 (Remarks to the Author)

This paper identifies a novel mechanism behind axon regeneration in the spinal cord of zebrafish. It is a very thorough and nicely described study that makes a significant advance in a field of general interest.

1. The question of whether cells precede axons is an old one, which always generates controversy, so it would be good if this could be a definitive answer. The authors should tell us whether the glial labels that they have used will identify all the process of all glial cells. It could be that the axons are growing on other cell types. A stain for a general cytoplasmic/cytoskeletal marker to show whether there are axons not in contact with any cell type would be a good addition.
2. There are a large number of reporters and interventions in the paper, and it is easy to lose track. A table with some description – which would probably have be supplementary for space reasons- would be a useful addition.
3. The various reporters etc. are of varying selectivity. For instance TCF can be activated by a variety of stimuli while axin is more specific. I think that they authors have done the correct experiments to demonstrate specificity, but some comment on the strategy would help.
4. There is an active debate in the mammalian regeneration field about the origin of cells that populate spinal injuries- are they meningeal, perivascular etc. Is there a zebrafish equivalent of the pericyte that could be involved in the injury response in zebrafish?
5. Why did the authors choose to study Col XII instead of Col VI?
6. Page 13. In the final paragraph we suddenly change to inhibition of Wnt signalling using dkk1, while the rest of the paper uses axin. It is better to be consistent if possible- was the axin approach not successful?
7. The various interventions do not stop regeneration completely, but reduce it by 40%. What might be responsible for the remaining regeneration? How does the present story fit with the previous tenascin story?
8. What might the receptor for Col XII be?

Reviewers' comments:

Reviewer #1 (Remarks to the Author):

This manuscript by Wehner, D. et al. showed that Wnt signal-dependent Collagen XII expression is necessary for axonal regeneration. Using zebrafish larvae the authors found that spinal cord transection induces Wnt signal activity in fibroblast-like cells and the activity is pivotal for axonal growth and functional recovery. Wnt signal increased the expression of Collagen XII in the lesion site, and its downregulation suppressed axonal growth and functional recovery. Taken together with the observation that glial bridges were not an obligatory substrate for dynamic axonal growth, the authors concluded that the lesion site environment regulated by Wnt signaling is important for regeneration of injured spine.

Experimental designs are elegant, and the obtained results are solid. As the authors described, Wnt signaling is known to control fibroblast biology and ECM deposition, and the lesion site ECM promotes spinal cord regeneration in fish. Therefore, the novelty in this study was to identify Collagen XII as an Wnt signaling downstream molecule and to show that Collagen XII is necessary for axonal regeneration by loss of function experiments.

The following experiments are required to strengthen the authors' conclusion.

1. In Fig 1A-C, the authors showed that initially formed axonal and glial fascicles are independently distributed. In addition, axonal connections can be formed independently of glial connection. From these observation, they claim that axonal growth cones extend into lesion sites independently of glial process formation. However, it is not clear whether axonal growth occurs independently of astrocytes per se. The authors can address this issue by depleting astrocytes.

RESPONSE: This is a very interesting suggestion. Using a transgenic line in which we can selectively ablate GFAP⁺ glia cells (generously provided by Prof. Didier Stainier; Matsuoka et al., 2016. Elife) we find a strong depletion of glial processes in the lesion site. However, the proportion of larvae with crossed axons after spinal cord lesion is not reduced, supporting a relatively minor role of GFAP⁺ glial cells in supporting axonal regeneration in zebrafish larvae. This agrees with our previous time-lapse video microscopy observations of glia-independent growth of axons. These data have been added as Fig. 1d-e to the revised manuscript and are highlighted on page 6-7.

2. The authors need to provide more details on exactly how images were scored in Fig. 1C.

RESPONSE: We agree with the reviewer that more detail is needed. We added a more detailed explanation to the Material and Methods section on page 31, clearly setting out our criteria to score animals as having bridged axonal/glial labelling or not. In Fig. 1c we did not correlate glial and axonal labelling in the same animals. However, glial bridges without axonal bridges were never observed.

3. *The authors claim “The majority of fascicles that entered the lesion site were composed of only axons (49-53%), some fascicles contained both glial and axonal processes (20-21%) and some pure glial fascicles were observed (27-30%)”. However, it is not clear how they count neural and glial fascicles.*

RESPONSE: We agree that more description is needed. We added a more detailed explanation to the Material and Methods section, which can be found on page 31-32 of the revised manuscript and we added information on fascicle numbers analysed to the Results.

4. *The authors showed that col1a2 and fin1a are expressed in the lesion site in Fig. 1D and that the expression levels of these mRNAs are not reduced by IWR-1 treatment. To address the involvement of Collagen I and Fibronectin 1 in axonal regeneration, the effect of reduction of these proteins by MO should be tested.*

RESPONSE: Analysing the function of other ECM molecules would be interesting. However, we find that genes coding for Col XII but not Col I or Fibronectin to be controlled by the Wnt/ β -catenin pathway (see Fig. 4a). As we analyse Wnt-dependent changes in the ECM composition of the spinal lesion site in this study, we believe that such experiments would go beyond the scope of this study. Manipulating these ECM factors would not provide further insight into the mechanism by which Wnt/ β -catenin signalling controls functional axon regeneration. However, we present new experimental data showing that overexpression of *col12a1a* can rescue effects of Wnt/ β -catenin pathway inhibition (Fig. 7a-d, Supplementary Fig. 13a-b) and is sufficient to accelerate regeneration (Fig. 7e). This identifies Col XII as a major pro-regenerative component of the lesion site ECM. We provide an explanation in the Discussion section of the revised manuscript on page 22 that Col I and Fibronectin are also likely to contribute.

5. *The important finding of this study is that Wnt signal-induced Collagen XII is required for spinal cord regeneration. Is col12a1a/b a direct target gene of Wnt signaling? If so, the authors should examine whether TCF/LEF binds to the col12a1a/b promoter region by ChIP assay and perform its promoter analysis. Otherwise, the underlying mechanism that Collagen XII expression is induced by Wnt signaling must be clarified.*

RESPONSE: We agree that it would be interesting to gain more insight into how Wnt controls *col12a1a/b* transcription. We present new data in the revised manuscript that induction of *col12a1a/b* lags behind that of the directly regulated Wnt target gene *axin2* (Supplementary Fig. 8e) and that Wnt/ β -catenin signalling does not have to be sustained for *col12a1a/b* expression (Supplementary Fig. 8d). These data strongly suggests that regulation of *col12a1a/b* by Wnt/ β -catenin signalling is likely indirect. We discuss these results in the revised manuscript on page 22-23. Unfortunately, we are unable to do ChIP, because of the lack of commercially available zebrafish-specific Tcf/Lef antibodies.

6. *Co-localization of axonal processes and ECM shown in Fig. 1E and Fig. 4G is partial. It is hard to understand axons navigate along ECM. More precise description*

and quantification are required.

RESPONSE: We agree that a better description of axon/ECM interactions will strengthen the manuscript. In the revised manuscript, we now give additional descriptions of the axon-ECM relationship in the Results section (page 7 and page 16) and add comprehensive 3D-views of axons and ECM as Supplementary Movies (Supplementary Movies 3-5). Additionally, in the Material and Methods section we now give a more detailed description of our quantification process (page 32).

7. In Fig. 4G the time course of appearance of disappearance of col12a1a/b after injury should be shown. Is the association between newly formed axon and Collagen XII observed at early time point such as at 0.5 dpl? Are 6 x TCF:dGFP-expressing cells present in the lesion site at 0.5 dpl? One would like to know whether the close contact of spinal axon and Collagen XII is important for regeneration.

RESPONSE: We are grateful for this helpful suggestion. We have generated the requested data. At 0.5 dpl, incipient Wnt reporter activity (Supplementary Fig. 2c) as well as *col12a1a/b* mRNA (Fig. 4b) is detectable, but axon regrowth (anti-acetylated Tubulin) and Col XII protein accumulation is observed only by 1 dpl (Supplementary Fig. 1a, 9c). Hence, the spatiotemporal correlation between Wnt/ β -catenin pathway activity, axon regrowth and *col12a1a/b* gene expression and Col XII deposition is confirmed. We state these new observations in the revised manuscript on page 5, page 8, page 14, and page 16.

8. The authors suggested that Wnt signal may not have to be sustained for transcription of col12a1a/b in Fig. 4E. The authors need to confirm that IWR-1 does not affect col12a1a/b expression after 1 dpl.

RESPONSE: This is another very good point. We have performed this experiment. The data, which can be found in Supplementary Fig. 8d of the revised manuscript, show that blocking Wnt/ β -catenin signalling from 1 dpl (50% of the regeneration time) had no effect on *col12a1a/b* gene expression when analysed at 2 dpl. This confirms that Wnt/ β -catenin signalling does not need to be sustained for *col12a1a/b* expression. These findings are described on page 14.

9. Figs 5 and 6 demonstrated that col12a1a/b expression is required for axonal regeneration using MO and CRISPR/cas9. Gain of function experiments are required to determine the functional importance of Collagen XII. Does overexpression of Collagen XII promote axonal regeneration? Is it possible to rescue the phenotypes of col12a1a/b-depleted larvae by expression of col12a1a/b?

RESPONSE: We agree that gain-of-function experiments will significantly strengthen our conclusions. To this end, we have generated a stable transgenic fish line, in which we can conditionally overexpress *col12a1a* (Fig. 7a-b, Supplementary Fig. 13a-b). Using this fish, we now show that overexpression of *col12a1a* is sufficient to accelerate axonal regeneration (Fig. 7e). Importantly, we demonstrate that *col12a1a* overexpression rescues axon regeneration after Wnt/ β -catenin pathway inhibition to almost wildtype levels and swimming function is also significantly improved (Fig. 7a-

d). This demonstrates that *col12a1a* is sufficient to promote axonal regeneration and is a major downstream effector of the Wnt/ β -catenin pathway in zebrafish spinal cord regeneration. These data is described in the revised manuscript on page 18-20 and figure 7.

We aimed to rescue axonal regeneration after depletion of *col12a1a/b*. To that end we designed a new splice site Vivo-Morpholino (MO) against *col12a1a*, because the MO used (as well as the CRISPR/Cas9 manipulations) would target the *col12a1a* rescue transcript (and transgene respectively). However, the new splice-site directed MO did not efficiently target *col12a1a* (see image below), such that rescue experiments of direct *col12a1a/b* manipulations would be very difficult to achieve. We propose that the new rescue experiment of Wnt/ β -catenin pathway inhibition by *col12a1a* (see above) essentially presents a rescue by the *col12a1a* transgene in a situation of depleted *col12a1a/b*, and thus confirms the pivotal role of *col12a1a/b* for regeneration.

Fig. Microinjection of *col12a1a* splice MO into the zygote shows little efficiency in altering pre-mRNA splicing when analysed at 1 dpf.

Reviewer #2 (Remarks to the Author):

In this manuscript by Wehner et al the authors investigate the role of Wnt/b-cat signaling and Col12a in CNS regeneration. The authors demonstrate that following spinal cord transection in zebrafish larvae Wnt/b-cat signaling is up-regulated in fibroblast like cells at the lesion site, and that blocking Wnt/b-cat signaling greatly reduces spinal cord regeneration. The authors then identify Col12a as a downstream component of Wnt/b-cat signaling, and provide evidence that like Wnt/b-cat signaling Col12a is also required in vivo for spinal cord regeneration. Overall, this is a very nice study that takes full advantage of the zebrafish as a model to identify genes that promote spinal cord regeneration, providing molecular insights into this process currently not feasible in mammals. Most claims (see below) are supported by solid experimental data, often using several approaches/ reagents to confirm findings. Finally, this study will be of great interest to the field of CNS regeneration, including mammalian CNS regeneration as it identifies a transcriptional target for Wnt/b-cat signaling, Col12a, that promotes spinal cord regeneration.

Main Points

1. From their live cell imaging the authors conclude “that glial bridges are not an obligatory substrate for axon growth across the non-neural lesion site.” Given the longstanding debate in the field whether axons or glia first cross the injury site, the authors need to substantiate their initial findings. Glial processes just like axonal growth cones can be of very small diameter and hence hard to detect when cytoplasmic GFP is used. Thus, to substantiate their findings that glial bridges are not an obligatory substrate it appears imperative to conduct this experiment using membrane tagged markers for glia processes, or electron microscopy. Otherwise this statement should be softened to reflect the possibility that while not detected here glial process might well be present when regenerating axons traverse the injury site.

RESPONSE: We agree that these interactions need to be analysed in more detail. To functionally address the contribution of glial processes to regeneration, we added a experiments in which we depleted GFAP⁺ glial cells using the Nitroreductase system (please also see response 1 to Reviewer #1). This did not have an effect on axon regeneration (Fig. 1d-e), in contrast to reducing Col XII deposition (Fig. 5). This supports a relatively minor role of the glia as anticipated from our earlier time-lapse video microscopy analyses. However, we agree with the reviewer that we cannot exclude the possibility that we missed glial processes and we explicitly state this in the Discussion section in the revised manuscript on page 23. Additionally, we moderated our statements regarding glial guidance throughout the manuscript.

2. The authors examine expression of 44 zebrafish collagens after spinal cord injury and report that only Col6 and Col12a (1a, 1b) are down regulated when Wnt/b-cat signaling is inhibited. It was less clear which of the 42 collagens were up-regulated after spinal cord injury (without IWR-1 treatment), and specifically whether Col12(1a, 1b) are indeed up-regulated following injury.

RESPONSE: We are grateful for this comment. We now show unlesioned control animals in the time course for *col12a1a/b* expression (please also see response 7 to reviewer #1; Fig. 4b), clearly showing upregulation of gene expression. In the in situ hybridization screen, we used adjacent non-lesioned trunk tissue in the same larvae

as an internal negative control for the highly localised accumulation of mRNAs in the lesion site. Of note, our in situ hybridisation protocol is sufficiently sensitive to detect transcripts deep in the tissue of non-lesioned larvae at that age, as demonstrated in Supplementary Fig. 2j. We explain this now in the revised manuscript in the Results section on page 13 and in the Material and Methods section on page 34.

3. The authors provide compelling evidence that Col12a/b is up-regulated in a subset of Wnt/b-cat responsive cells and that forced Wnt8 expression (via hs:wnt8) enhances Col12a (1a, 1b) expression levels at the lesion site. Combined with the observation that reducing Col12a function recapitulates the phenotype (i.e. reduced regeneration) observed when blocking Wnt/b-cat signaling, this suggests that Wnt/b-cat signaling is required to activate Col12a (1a, 1b) expression. It therefore seems important to determine whether Col12a expression is sufficient to restore spinal cord regeneration in larvae in which Wnt/b-cat signaling is blocked. This 'rescue' experiment would complement the evidence for Wnt/b-cat inducing Col12 (a, b) expression gained mostly by overexpression (e.g. Axin, wnt8) analysis, and would provide some mechanistic insights into whether Col12A (1a, 1b) are the major Wnt/b-cat targets relevant for spinal cord regeneration.

RESPONSE: We agree with the reviewer that this is a crucial experiment adding mechanistic insight. We have generated a stable transgenic fish line, in which we can conditionally overexpress *col12a1a* (Fig. 7a-b, Supplementary Fig. 13a-b). Using this fish, we now show an almost complete rescue of axonal regeneration caused by Wnt/ β -catenin pathway inhibition. Moreover, overexpression of *col12a1a* alone is sufficient to accelerate spinal cord regeneration (Fig. 7e). These data identify Col XII as a major downstream effector of Wnt/ β -catenin signaling in the regenerating zebrafish spinal cord. We describe these data in the revised manuscript on page 18-20 and figure 7.

4. The authors show that Wnt/b-cat signaling is important in vivo for spinal cord regeneration. However, which step during the complex process of regeneration Wnt/b-cat signaling promotes remains unclear. For example, does Wnt/b-cat signaling initiate and/or sustain regeneration? By inducing Axin expression or starting IWR-1 treatment later and later time points post injury, it should be possible to determine the first time point when Wnt/b-cat signaling is required (i.e. the first time point at which blocking Wnt/b-cat signaling no longer affects regeneration). This additional data would provide important mechanistic insights into the role of Wnt/b-cat signaling in spinal cord regeneration.

RESPONSE: We thank the reviewer for this helpful suggestion. We have performed this experiment (Supplementary Fig. 5d). It shows that Wnt/ β -catenin signalling is critical between 12 and 24 hours post-lesion, consistent with a role in initiating *col12a1a/b* expression. These data are described in the revised manuscript on page 12. Additional new data show negligible axon regrowth and Col XII deposition at 12 hpl (Supplementary Fig. 1a, 9c). Hence, the temporal requirement of Wnt/ β -catenin signaling correlates well with axon regrowth starting between 12 and 24 hours after the injury.

5. Finally, without stable *Col12a* (1a, 1b) mutant lines it seems important to correlate the CRISP analysis on a larvae by larvae basis with the regeneration assay. Specifically, on a larvae by larvae basis what is the extent of regeneration with none, only *col12a*, only *col12b* or *col12a/b* bi-allelic inactivation?

RESPONSE: We agree that single larva analysis will strengthen the transient CRISPR manipulation experiments and we added this analysis. Single CRISPR/Cas9 manipulation of either *col12a1a* or *col12a1b* did not result in a detectable phenotype, possibly due to mutual compensation of the two paralogs, which both code for the Col XII alpha-chain (incorporated as Supplementary Fig. 12a and described on page 18 in the revised manuscript). Instead, we performed RFLP analysis in 169 individual larvae with *col12a1a* and *col12a1b* CRISPR/Cas9 manipulation. This showed significant correlation of the effectiveness of CRISPR/Cas9 manipulation of both *col12a1a* and *col12a1b* with the phenotype (axonal bridging). This represents an additional control for the specificity of the acute CRISPR manipulation. We incorporated this data as Supplementary Fig. 12b into the revised manuscript and describe it on page 18.

Reviewer #3 (Remarks to the Author):

This paper identifies a novel mechanism behind axon regeneration in the spinal cord of zebrafish. It is a very thorough and nicely described study that makes a significant advance in a field of general interest.

1. The question of whether cells precede axons is an old one, which always generates controversy, so it would be good if this could be a definitive answer. The authors should tell us whether the glial labels that they have used will identify all the process of all glial cells. It could be that the axons are growing on other cell types. A stain for a general cytoplasmic/cytoskeletal marker to show whether there are axons not in contact with any cell type would be a good addition.

RESPONSE: This is an important point, also raised by the other reviewers. We decided to take a functional approach to further address the contribution of glial cells by genetically ablating the same. This did not have an effect on axon regeneration, which further supports a rather minor role of glia in axon regeneration (Fig. 1d-e and described on page 6-7; please also see our response to reviewer #2, point1). In addition we have moderated our statements about the glia throughout the manuscript to reflect the possibility that observation and ablation may have missed some glial processes.

2. There are a large number of reporters and interventions in the paper, and it is easy to lose track. A table with some description – which would probably have been supplementary for space reasons- would be a useful addition.

RESPONSE: We thank the reviewer for this helpful comment. We added such a table as Supplementary Table 1 (see Supplemental Information, page 24-26) to the revised manuscript.

3. The various reporters etc. are of varying selectivity. For instance TCF can be activated by a variety of stimuli while axin is more specific. I think that they authors have done the correct experiments to demonstrate specificity, but some comment on the strategy would help.

RESPONSE: We agree that more description of our controls is helpful. Our new Supplementary Table 1 also contains descriptions of the mechanisms of actions for the different methods of Wnt/ β -catenin pathway inhibition. We also tested specificity of the reporter (trans)genes used in our system by demonstrating disappearance of labelling after inhibition of Wnt/ β -catenin signalling in (Supplementary Figs. 2f-g, 8e).

4. There is an active debate in the mammalian regeneration field about the origin of cells that populate spinal injuries- are they meningeal, pervascular etc. Is there a zebrafish equivalent of the pericyte that could be involved in the injury response in zebrafish?

RESPONSE: We thank the reviewer for raising this interesting point. We have now used a transgenic reporter fish in which GFP is driven by the *pdgrb*. This is a marker

for pericytes in uninjured zebrafish (Ando et al. 2016. Development). After a lesion we find pronounced appearance of *pdgfrb*:GFP⁺ in the lesion site and that the *pdgfrb*:GFP transgene marks *col12a1a/b*-expressing cells in the lesion site. This points to a potential pericyte origin of lesion site cells as described in spinal cord-lesioned mammals (Goritz et al., 2011. Science). Additional new experiments show that Wnt/ β -catenin pathway inhibition does not prevent appearance of these cells, such that the effect of Wnt/ β -catenin signaling is unlikely on cell migration. We have added this information as Supplementary Fig. 8f-h to the revised manuscript and describe it on page 15.

5. *Why did the authors choose to study Col XII instead of Col VI?*

RESPONSE: We agree that giving our reasons for focussing on ColXII are important. We analysed Col XII in detail, because both *col12a1* paralogs that code for the alpha chain of the protein are under transcriptional control of Wnt/ β -catenin signalling. Consequently, Wnt/ β -catenin pathway inhibition leads to reduced presence of Col XII protein. For Col VI only one of three tested paralogs is controlled by the Wnt/ β -catenin pathway. Thus it is possible that significant levels of functional Col VI are still made under condition of Wnt/ β -catenin pathway inhibition. We now give this reason in the manuscript. Importantly, our new experiments show that overexpression of *col12a1a* leads to an almost complete rescue of axonal regeneration in fish in which the Wnt/ β -catenin pathway was inhibited. This confirms that Col XII is indeed a major downstream effector of the Wnt pathway (Fig. 7a-d, Supplementary Fig. 13a-b). We discuss a possible contribution of Col VI in the Results and Discussion section of the revised manuscript on page 13 and page 21.

6. *Page 13. In the final paragraph we suddenly change to inhibition of Wnt signalling using dkk1, while the rest of the paper uses axin. It is better to be consistent if possible- was the axin approach not successful?*

RESPONSE: This is a helpful comment. We apologize that we did not sufficiently make clear that *dkk1* over-expression experiments were not stand-alone, but rather represent important controls for our other Wnt pathway manipulations using IWR-1 treatment and *axin1* overexpression. Effects of *dkk1* over-expression on axonal regeneration and Col XII deposition are also seen by IWR-1 treatment and *axin1* overexpression (Supplementary Fig. 5, 9b; Fig. 4h). *dkk1* overexpression is an important control, because it inhibits Wnt/ β -catenin signalling with great specificity at the receptor level, whereas Axin1 and IWR-1 promote degradation of the transcriptional co-activator β -catenin. We explicitly state this now in the revised manuscript on page 11-12 and also mention this in the new Supplementary Table 1 that explains our transgenic tools.

7. *The various interventions do not stop regeneration completely, but reduce it by 40%. What might be responsible for the remaining regeneration? How does the present story fit with the previous tenascin story?*

RESPONSE: We agree that discussion of additional mechanisms of regeneration is warranted. We now express values as % of lesioned control to better reflect the

magnitude of the effects. Indeed our strongest manipulation with IWR-1 reduced the proportion of animals with axonal bridges by 72% (Supplementary Fig. 5a). Nevertheless, the spinal lesion site ECM is highly complex and other regeneration-promoting molecules, such as Tenascin or Fibronectin, which is not under the control of the Wnt/ β -catenin pathway, are also present. These may be responsible for the residual regeneration we observed. We now discuss this point in the Discussion section of the revised manuscript on page 22.

8. What might the receptor for Col XII be?

RESPONSE: We agree that potential receptor interactions with ColXII would be interesting to determine. Presently, we have no new data to answer that question. Col XII has been described as an organiser of other ECM components, such that its effect on axon growth may not be receptor mediated. Alternatively, integrin type receptors may bind Col XII directly through its NC1 domain as shown for Col IV. We have expanded on the discussion of these points in the revised manuscript on page 21-22.

Reviewers' Comments:

Reviewer #1:

Remarks to the Author:

The authors performed several experiments according to my comments. I believe that the revised manuscript is improved.

Reviewer #2:

Remarks to the Author:

In this revised version, the authors have addressed all of my concerns. They should be commended for being very receptive to all reviewers suggestions and for addressing most comments experimentally.

Reviewer #3:

Remarks to the Author:

The three referees made constructive suggestions, including recommendations for further experiments. The authors have done a considerable amount of new work, all of which supports the ideas in the paper. This is now a large and convincing body of work. I do not have any further suggestions, and I do not think that another round of reviewing and additions to the paper will improve it.

Response to reviewers:

REVIEWERS' COMMENTS:

Reviewer #1 (Remarks to the Author):

The authors performed several experiments according to my comments. I believe that the revised manuscript is improved.

Reviewer #2 (Remarks to the Author):

In this revised version, the authors have addressed all of my concerns. They should be commended for being very receptive to all reviewers suggestions and for addressing most comments experimentally.

Reviewer #3 (Remarks to the Author):

The three referees made constructive suggestions, including recommendations for further experiments. The authors have done a considerable amount of new work, all of which supports the ideas in the paper. This is now a large and convincing body of work. I do not have any further suggestions, and I do not think that another round of reviewing and additions to the paper will improve it.

RESPONSE: We are grateful for the insightful comments of the reviewers, which helped us to improve the manuscript.